# Neoadjuvant Treatment for Locally Advanced Rectal Cancer: Current Status and Future Directions

**DOI:** 10.3390/cancers17152540

**Published:** 2025-07-31

**Authors:** Masayoshi Iwamoto, Kazuki Ueda, Junichiro Kawamura

**Affiliations:** Department of Surgery, Kindai University Faculty of Medicine, 377-2, Ohnohigashi, Osaka-Sayama 589-8511, Osaka, Japan; ueda-k@med.kindai.ac.jp (K.U.); kawamuraj@med.kindai.ac.jp (J.K.)

**Keywords:** locally advanced rectal cancer, neoadjuvant therapy, chemoradiotherapy, total neoadjuvant therapy, non-operative management, quality of life, organ preservation, molecular biomarkers, personalized medicine, artificial intelligence

## Abstract

Locally advanced rectal cancer is a serious disease that often requires complex treatment to prevent recurrence and improve survival. Traditionally, a combination of surgery, radiation, and chemotherapy has been used, but new strategies, such as total neoadjuvant therapy, which involves administering both chemotherapy and radiotherapy prior to surgery, and non-operative management for patients who achieve a complete clinical response, are expanding options. These approaches not only improve oncological outcomes but also aim to preserve patients’ quality of life. This review summarizes the latest approaches to preoperative treatment for rectal cancer, highlights the importance of tailoring therapy to each individual, and discusses promising developments like molecular biomarkers and artificial intelligence. By integrating these innovations, we move closer to truly personalized care that optimizes both survival and well-being for people facing rectal cancer.

## 1. Introduction

Colorectal cancer (CRC) is a malignancy with high incidence and mortality rates worldwide, especially in countries experiencing economic development. In 2022, approximately 1.93 million new CRC cases were diagnosed worldwide, making it the third most common cancer and the second leading cause of cancer-related death [1]. Rectal cancer accounts for about 30% of all CRCs, and its distinctive anatomical location and clinical characteristics often necessitate different treatment strategies from those for colon cancer. Particularly, due to anatomical constraints within the pelvis and proximity to adjacent organs, patients with locally advanced rectal cancer (LARC) carry a high risk of local recurrence, making local control a significant clinical challenge [2].

Total mesorectal excision (TME), the standard surgical technique for rectal cancer, is crucial for achieving optimal local control and has been widely adopted over the past few decades [3,4,5]. However, in advanced rectal cancer, the local recurrence rate after surgery alone remains unacceptably high [6]. To address this issue, preoperative radiotherapy (RT) or chemoradiotherapy (CRT) was introduced in the 1980s—initially in Western countries—with the aim of improving local control, and these modalities are now firmly established components of standard care [7].

More recently, efforts to further optimize preoperative management have led to the concepts of total neoadjuvant therapy (TNT), which incorporates both RT and systemic chemotherapy before surgery, and non-operative management (NOM) for patients who achieve a clinical complete response (cCR). These strategies have increasingly been investigated in clinical trials and incorporated into treatment guidelines [2,8].

This narrative review aims to provide an overview of the current status of preoperative treatment for LARC, discuss unresolved issues, and outline future directions for treatment strategies.

## 2. Literature Search Strategy

In order to provide a comprehensive and up-to-date overview of neoadjuvant treatment strategies for LARC, we conducted a structured literature search of the PubMed, Embase, and Web of Science databases for studies published in English up to June 2025. The search strategy employed combinations of key terms, including “rectal cancer,” “locally advanced rectal cancer,” “neoadjuvant therapy,” “radiotherapy,” “chemoradiotherapy,” “total neoadjuvant therapy,” “watch and wait,” “non-operative management,” “immunotherapy,” “targeted therapy,” and “molecular classification”. We included randomized controlled trials (RCTs), meta-analyses, major cohort studies, clinical guidelines, and narrative reviews focusing on neoadjuvant RT, CRT, TNT, NOM, neoadjuvant chemotherapy (NAC), targeted therapies, and immunotherapies in adult patients with LARC. Preference was given to studies with the highest level of evidence and the greatest clinical relevance. International clinical guidelines, such as those from the National Comprehensive Cancer Network (NCCN) and the European Society for Medical Oncology (ESMO), were also reviewed. Studies involving pediatric populations, non-primary rectal cancer, small case series with fewer than 10 patients, conference abstracts, non-English articles, and preclinical or in vitro studies were excluded. The initial screening of titles and abstracts was performed by the first author, followed by a full-text review. Selected references were further evaluated by all co-authors for clinical importance, originality, and representativeness. Discrepancies were resolved by consensus. Literature selection and data synthesis were conducted in accordance with the PRISMA 2020 guidelines; however, a flow diagram is not presented owing to the narrative design of this review [9].

## 3. Definition, Diagnosis, and Molecular Classification of LARC

### 3.1. Definition of LARC

LARC is generally defined as a subgroup of rectal cancers that invade beyond the muscularis propria and/or involve regional lymph nodes, but without distant metastasis. According to the TNM classification, LARC corresponds clinically to Stages II or III, encompassing T3 tumors with any N, T1–2 tumors with N1–2, and T4 tumors with any N, all without distant metastasis (M0). Anatomically, rectal cancer is defined as a tumor with its distal margin located within 15 cm from the anal verge, a criterion that is widely adopted in both the ESMO and the NCCN guidelines [2,8]. Pathologically, most rectal cancers are adenocarcinomas, with LARC frequently discussed in the context of high-risk cases eligible for preoperative treatment.

The term “LARC” is a crucial clinical concept that clearly denotes a group of patients at high risk of recurrence, playing a vital role in treatment strategy planning and prognostic assessment. In recent years, it has become recognized that not only simple TNM classification but also multiple local extension factors—such as extramural vascular invasion (EMVI) and the distance to the mesorectal fascia (MRF)—influence treatment decisions, highlighting the importance of individualized risk assessment.

### 3.2. Diagnosis of LARC

The accurate evaluation of local tumor extension and lymph node involvement is essential in diagnosing LARC. Precise clinical staging directly informs preoperative treatment eligibility, surgical planning, and prognosis stratification, emphasizing the need for high-quality diagnostic methods [2,8].

For T staging, endorectal ultrasonography (ERUS) is particularly valuable in early-stage lesions (T1–T2) because it offers accurate differentiation of submucosal invasion and muscularis propria involvement [10]. However, for advanced lesions (T3–T4), high-resolution magnetic-resonance imaging (MRI) is currently considered the gold standard [2,8].

The MERCURY trial first demonstrated that a standardized high-resolution MRI protocol allows for a precise preoperative assessment of extramural depth of tumor spread and the circumferential resection margin (CRM; distance between the tumor and the MRF). This study established the clinical significance of preoperative MRI assessment internationally, showing a high concordance rate between MRI-based CRM measurements and pathological findings. Moreover, cases with MRI-based CRM less than 1 mm were associated with poorer outcomes in terms of local recurrence, overall survival (OS), and disease-free survival (DFS) [11,12]. Thus, MRI-based preoperative evaluation has become indispensable for optimizing treatment strategies and predicting prognosis beyond traditional TNM classification.

MRI also plays an important role in the assessment of lymph node metastasis. It enables the identification of not only perirectal nodes but also lateral pelvic sidewall lymph nodes, providing crucial information for evaluating the risk of local recurrence. However, the assessment of nodal malignancy with MRI requires a combination of imaging criteria, including size, morphology, and signal characteristics, and current diagnostic performance remains limited. In particular, differentiating reactive enlargement from metastasis and detecting micrometastases are challenging; thus, MRI alone cannot reliably determine the presence of nodal involvement [10]. Furthermore, even when combining ERUS, computed tomography (CT), and MRI, meta-analyses and large-scale studies have shown that the reliability of nodal assessment is limited, and comprehensive evaluation with other clinical information is currently recommended [13,14].

Improving the accuracy of risk stratification also remains an important issue. While many clinical trials and guidelines treat T3 and T4 as a single group, in clinical practice, factors such as the depth of extramural invasion in T3 tumors, distance to the MRF, and presence of EMVI have a significant impact on prognosis [15,16,17,18]. Unlike CT, MRI can visualize these detailed risk factors, playing a crucial role in personalized treatment. Furthermore, numerous studies, including the MERCURY trial, emphasize the importance of standardized imaging techniques, expert radiological interpretation, and multidisciplinary team evaluations for enhancing diagnostic accuracy and treatment outcomes [19].

### 3.3. Molecular Classification

In recent years, the significance of molecular classification, alongside traditional pathological and anatomical classifications, has markedly increased in decision-making for CRC treatment [20,21]. Advances in omics technologies have enabled the comprehensive analysis of cancer genomes and epigenomes, making molecular subtyping increasingly valuable for prognostication, treatment planning, and the selection of targeted therapies and immunotherapy [21].

The molecular classification of CRC has been based on a variety of mechanisms, including genomic and epigenomic alterations, microsatellite instability (MSI), DNA mismatch repair (MMR) status, the WNT pathway, and RAS/RAF/MEK signaling [20]. However, outcomes from conventional gene-expression-based classification systems lack consistency. To overcome this issue, the international consortium, organized by European and US researchers, conducted large-scale collaborative efforts, integrating existing data to propose the “Consensus Molecular Subtypes (CMS)” [22].

The CMS classification divides CRC into four major molecular subtypes, each with distinct biological characteristics, prognosis, and therapeutic responsiveness. CMS1 (MSI immune, ~14%) is characterized by high MSI due to MMR deficiency and frequent BRAF mutations, resulting in abundant neoantigen expression, significant intratumoral immune cell infiltration, and the pronounced activation of immune evasion pathways. CMS2 (canonical, ~37%) demonstrates epithelial gene expression patterns and the significant activation of WNT and MYC signaling. CMS3 (metabolic, ~13%) is notable for a high frequency of KRAS mutations and gene expression reflecting metabolic pathway abnormalities. CMS4 (mesenchymal, ~23%) exhibits mesenchymal features, with activation of TGF-β signaling, epithelial–mesenchymal transition (EMT), increased fibrosis and angiogenesis, and a tendency to form an inflammatory and immunosuppressive microenvironment. This subtype is associated with poor prognosis and high resistance to chemotherapy and targeted therapies. Furthermore, following the establishment of the CMS system, new molecular classifications have been proposed, such as the “Single-cell intrinsic CMS (iCMS),” which enables subclassification at the epithelial cell level using single-cell RNA sequencing, and the “Pathway-derived subtypes (PDS),” which categorizes tumors based on pathway-level clustering rather than conventional gene expression data [23,24].

Clinically, molecular biomarkers—including KRAS/NRAS/BRAF mutations, HER2 amplification, PIK3CA mutations, MSI/MMR status, and the tumor microenvironment—play an important role in therapeutic decision-making [20]. Immune checkpoint inhibitors (ICI) are highly effective in tumors that exhibit MSI-high or MMR-deficient status, while anti-EGFR antibody therapy is indicated for RAS/BRAF wild-type tumors [25]. However, the observation that many patients with KRAS wild-type CRC do not respond to cetuximab or panitumumab suggests the presence of additional resistance mechanisms, highlighting the limitations of single-gene targeted approaches. Moving forward, it is expected that integrating multi-layered information—such as multiple gene mutations, pathway activities, and tumor/stromal/immune subtypes—will further optimize individualized treatment for CRC [21].

## 4. Radiotherapy/Chemoradiotherapy (RT/CRT)

Rectal cancers, especially those located in the lower rectum and anal canal, generally exhibit poorer oncological outcomes compared to colon cancers, partly due to a higher local recurrence rate [26]. Factors that may cause local recurrence after rectal cancer surgery include tumor extension into adjacent pelvic organs or tissues, lateral pelvic sidewall lymph node metastasis, technical difficulties of surgery in the confined pelvic space resulting in residual cancer, and possible intraoperative tumor seeding.

Surgical resection for the local recurrence of rectal cancer is highly invasive, technically challenging, often associated with serious complications, and has poor long-term outcomes [27,28,29]. When radical re-excision is not possible, patients may experience severe impairment of quality of life (QOL) due to intractable pelvic pain, bleeding from recurrent tumors, or abscess formation. Therefore, the avoidance of local recurrence is crucial in surgery for rectal cancer, and it is essential to ensure a CRM and achieve a complete TME [30].

Preoperative CRT for rectal cancer was introduced in the 1980s, at a time when the concepts of CRM and TME had not yet been established, with the intent to reduce local recurrence and improve survival. Although early-stage rectal cancer has been successfully managed with TME alone, the postoperative local recurrence rate in LARC patients remains unacceptably high, and CRT has become the standard preoperative modality to enhance local control (Table 1). From a theoretical perspective, preoperative therapy offers several advantages: the vascular architecture of the rectum remains intact, which improves tissue oxygenation, drug delivery, and radiosensitivity. Preoperative CRT may also eradicate peritumoral micrometastases and limit tumor cell dissemination or implantation, even in cases where TME is suboptimal. Additionally, unlike postoperative irradiation, preoperative radiotherapy is delivered prior to the development of small bowel adhesions, thereby minimizing high-dose exposure to the radiosensitive small intestine and improving treatment tolerability. Clinically, preoperative therapy can induce tumor downsizing and downstaging, leading to increased rates of R0 resection and improved opportunities for sphincter preservation.

### 4.1. Superiority of Preoperative RT over Surgery Alone

Multiple RCTs conducted in the 1980s demonstrated that preoperative RT significantly reduced the local recurrence rate compared to surgery alone in resectable rectal cancer [49,50,51,52]. Landmark studies—including the Stockholm trial (1980–1987; n = 849), Stockholm II trial (1987–1993; n = 557), and the Swedish Rectal Cancer Trial (1987–1990; n = 1168)—consistently showed that a preoperative short-course RT (SC-RT) regimen of 5 Gy administered over five fractions markedly decreased postoperative local recurrence relative to surgery alone [31,32,53,54,55]. Notably, in the Swedish trial, long-term follow-up over 13 years confirmed a sustained reduction in local recurrence rates (9% vs. 26%; *p* < 0.001) and a significant improvement in OS (38% vs. 30%; *p* = 0.008) in the RT group. However, it is important to acknowledge that these studies were performed prior to the widespread recognition of the oncological significance of TME. In the surgery-alone arms, the majority of resections were conducted using blunt dissection rather than the standardized TME technique. As a result, the local recurrence rates in the surgery-alone groups exceeded 20%, which necessitates cautious interpretation of these results in the context of current surgical standards.

The Dutch trial subsequently enrolled 1861 patients with resectable rectal cancer between 1996 and 1999 to compare the same SC-RT protocol with surgery alone under the strict requirement that all resections include TME [6,33,34]. At 10 years, preoperative SC-RT reduced the local recurrence rate to 5% versus 11% in the surgery-alone group (*p* < 0.0001). In contrast, the 10-year OS did not differ significantly between the two cohorts (48% vs. 49%, *p* = 0.86), indicating that the principal benefit of preoperative RT in the modern TME era lies in improved local control rather than in extended OS.

### 4.2. Comparison of Preoperative RT and Preoperative CRT

Two RCTs published in 2006 investigated whether the addition of chemotherapy to preoperative long-course radiotherapy (LC-RT) could improve treatment outcomes in rectal cancer.

The EORTC 22921 trial enrolled 1101 patients with resectable T3 or T4 rectal cancer between 1993 and 2003, randomly assigning them to four treatment arms: (i) preoperative LC-RT alone, (ii) preoperative long-course chemoradiotherapy (LC-CRT), (iii) preoperative LC-RT followed by adjuvant chemotherapy, and (iv) preoperative LC-CRT followed by adjuvant chemotherapy [35]. The CRT-containing arms demonstrated significantly greater tumor shrinkage and higher rates of pathological downstaging compared to the RT-alone arms. At 10 years, cumulative local recurrence risk was highest in the RT-alone group and significantly lower in the CRT and postoperative-chemotherapy arms (i, 22.4%; ii, 11.8%; iii, 14.5%; iv, 11.7%; *p* = 0.0017). However, the 10-year OS was nearly identical between the preoperative RT and CRT groups—49.4% (95% CI, 44.6–54.1) versus 50.7% (95% CI, 45.9–55.2), respectively (HR = 0.99, 95% CI, 0.83–1.18, *p* = 0.91)—and the 10-year DFS did not differ significantly between the two arms (44.2% vs. 46.4%; HR = 0.93, 95% CI, 0.79–1.10, *p* = 0.38) [36,37].

Similarly, the FFCD 9203 trial randomized 762 patients with cT3–T4 resectable rectal cancer (1993–2003) to receive either preoperative LC-RT alone or LC-CRT [38]. TME was recommended for all patients, and adjuvant 5-fluorouracil plus leucovorin (5-FU/LV) was administered post-operatively. The CRT group achieved a significantly higher pathological complete response (pCR) rate (11.4% vs. 3.6%, *p* < 0.0001) and a lower 5-year local recurrence rate (8.1% vs. 16.5%, *p* = 0.004). However, grade 3–4 acute toxicity was more common in the CRT group (14.9% vs. 2.9%, *p* < 0.0001), and the 5-year OS did not differ significantly between groups (67.4% vs. 67.9%, *p* = 0.684).

Collectively, these results indicate that the addition of chemotherapy to preoperative LC-RT improves local control but does not prolong the OS and is associated with a moderate increase in acute toxicity.

### 4.3. Comparison of Preoperative SC-RT and LC-CRT

The Polish trial randomly assigned 312 patients with resectable T3/4, N0–2 rectal cancer to receive either SC-RT or LC-CRT between 1999 and 2002 [39]. In the SC-RT arm, 25 Gy in five fractions was followed by surgery within 7 days, whereas the LC-CRT arm received 50.4 Gy in 28 fractions with concomitant 5-FU/LV, followed by surgery 4–6 weeks later. TME was mandatory in both groups. The primary endpoint—sphincter preservation—did not differ significantly between arms (LC-CRT, 61.2% vs. SC-RT, 58.0%; *p* = 0.570). However, the pCR rate was markedly higher with LC-CRT (16.1% vs. 0.7%; *p* < 0.001), and the incidence of positive CRM was lower (4.4% vs. 12.9%; *p* < 0.001). At a median follow-up of 48 months, no significant differences were observed in the OS (66.2% vs. 67.2%; *p* = 0.960), DFS (55.6% vs. 58.4%; *p* = 0.820), or local recurrence (15.6% vs. 10.6%; *p* = 0.210). Postoperative morbidity and late complications were comparable, but grade ≥ 3 acute toxicity was more frequent with LC-CRT (18.2% vs. 3.2%; *p* < 0.001).

The TROG trial enrolled 326 cT3 patients between 2001 and 2006, randomizing them to the same SC-RT or LC-CRT regimens [40]. With a median follow-up of 5.9 years, there were no significant differences in 3-year local recurrence rates (LC-CRT, 4.4% vs. SC-RT, 7.5%; *p* = 0.24), 5-year OS (70% vs. 74%; *p* = 0.62), and grade 3–4 late toxicity (8.2% vs. 5.8%; *p* = 0.53). Nonetheless, LC-CRT yielded higher pCR rates (15% vs. 1%; *p* < 0.001) and pathological down-staging rates (45% vs. 28%; *p* = 0.002).

A meta-analysis incorporating these two RCTs demonstrated that LC-CRT significantly improved pCR compared with SC-RT (risk ratio [RR] = 0.15, 95% CI 0.08–0.28; *p* < 0.00001) and showed a favorable trend toward greater tumor down-staging (RR = 0.61, 95% CI 0.37–1.01; *p* = 0.05) [56]. Conversely, LC-CRT was associated with a higher incidence of grade 3–4 acute toxicity (RR = 0.13, 95% CI 0.06–0.28; *p* < 0.00001). No significant differences were observed between the two modalities in R0 resection rate, sphincter preservation, local recurrence, distant metastasis, or OS.

Based on these findings, both SC-RT and LC-CRT are considered reasonable preoperative treatment options for resectable LARC, with generally comparable outcomes. SC-RT offers advantages such as reduced acute toxicity, lower cost, and shorter treatment duration, making it especially useful in centers with long waiting times or limited resources. However, for cases in which a high pCR rate is prioritized, LC-CRT may be preferred, although it should be noted that this does not necessarily translate to better long-term outcomes.

### 4.4. Superiority of Preoperative CRT Compared with Postoperative CRT

The German CAO/ARO/AIO-94 trial, reported in 2004, compared LC-CRT administered preoperatively with the same regimen delivered postoperatively in a randomized design [7,41]. Between 1994 and 2002, 823 patients with resectable cT3–4 or node-positive rectal cancer were randomized to either preoperative or postoperative CRT. The 10-year OS was comparable between the groups (59.6% vs. 59.9%; HR = 0.98, 95% CI 0.79–1.21, *p* = 0.85), as was the incidence of distant metastasis (29.8% vs. 29.6%; HR = 0.98, 95% CI 0.76–1.28, *p* = 0.91). In contrast, the 10-year local recurrence rate was significantly lower in the preoperative CRT group (7.1% vs. 10.1%; HR = 0.60, 95% CI 0.40–1.00, *p* = 0.048). Moreover, grade ≥ 3 acute and late toxicities were less frequent in the preoperative arm, underscoring its superior safety profile.

The MRC CR07/NCIC-CTG C016 trial randomized 1350 patients with resectable rectal cancer to preoperative SC-RT (25 Gy in 5 fractions) or selective postoperative CRT (45 Gy in 25 fractions plus 5-FU, administered only for patients with a positive CRM) [5,42]. The 3-year local recurrence rate, which was the primary endpoint, was significantly lower in the preoperative RT group (4.4% vs. 10.6%, HR = 0.39, *p* < 0.0001). The 3-year DFS was also superior in the preoperative RT group (77.5% vs. 71.5%, HR = 0.76, *p* = 0.013). There was no significant difference in the OS (80.3% vs. 78.6%, HR = 0.91, *p* = 0.40), and postoperative morbidity rates were similar in both groups, confirming the safety of the preoperative approach.

These results established preoperative CRT as the standard of care for LARC, as it improves local control and reduces toxicity compared to postoperative CRT, even though it does not improve OS.

### 4.5. Chemotherapy Regimens Combined with Preoperative CRT

Initially, intravenous 5-FU/LV was the standard chemotherapy regimen during preoperative CRT, requiring inpatient management throughout treatment. Subsequently, in terms of oral anticancer agents, 5-FU/LV has been replaced with capecitabine (Cape), which is now widely used in preoperative CRT. There have also been trials that have investigated the addition of oxaliplatin or irinotecan, which are standard agents in advanced/recurrent CRC, to further enhance tumor shrinkage and improve long-term outcomes.

The NSABP-R04 trial compared continuous intravenous 5-FU with Cape and assessed the effect of adding oxaliplatin in a multicenter phase III RCT [43,44]. Among 1608 enrolled patients, there were no significant differences between 5-FU and Cape in 3-year local recurrence rates (11.2% vs. 11.8%, HR = 1.0, *p* = 0.98), 5-year DFS (66.4% vs. 67.7%, HR = 0.97, *p* = 0.70), or 5-year OS (79.9% vs. 80.8%, HR = 0.94, *p* = 0.61). The addition of oxaliplatin also did not show significant benefits in terms of the local recurrence rate (11.2% vs. 12.1%, HR = 0.94, *p* = 0.70), DFS (69.2% vs. 64.2%, HR = 0.91, *p* = 0.34) or OS (81.3% vs. 79.0%, HR = 0.89, *p* = 0.38). Furthermore, the completion rate of treatment was lower, and grade 3 or higher diarrhea and severe toxicities were significantly more frequent in the oxaliplatin groups (*p* < 0.0001).

Several additional phase III trials evaluated oxaliplatin intensification. In the ACCORD 12 study, 598 patients with T3–4, M0 disease were randomized to receive RT (45 Gy/25 fr) plus Cape (Cap45 group) or RT (50 Gy/25 fr) plus Cape plus oxaliplatin (Capox50 group) [45,46]. While CRM positivity was lower in the Capox50 group (19.3% vs. 9.9%, *p* = 0.022), there was no significant difference in the pCR rate (13.9% vs. 19.2%, *p* = 0.09) or 5-year local recurrence, DFS, or OS. Grade 3–4 acute toxicity was significantly higher in the Capox50 group (25.4% vs. 10.9%, *p* < 0.001), raising issues with treatment completion.

Similarly, the STAR-01 trial randomized 747 patients with cT3–4 and/or cN1–2 rectal cancer to 5-FU plus RT or 5-FU plus oxaliplatin plus RT [57]. The primary endpoint, which was the pCR rate, did not differ (16% vs. 16%, *p* = 0.904), nor did pathological T-stage, nodal involvement, tumor regression grade, or CRM positivity. Conversely, grade 3–4 toxicity increased significantly with oxaliplatin (8% vs. 24%, *p* < 0.001), and adherence to both RT and chemotherapy declined.

By contrast, the CAO/ARO/AIO-04 trial randomized 1265 patients with cT3–4 or cN+ rectal cancer to standard 5-FU or oxaliplatin-containing CRT, using a modified oxaliplatin schedule to enhance tolerability [47,48]. Completion rates were high (RT 94%, chemotherapy 85%), and the oxaliplatin arm achieved significant improvements in the pCR rate (17% vs. 13%, *p* = 0.038) and 3-year DFS (75.9% vs. 71.2%, HR = 0.79, *p* = 0.03). However, grade ≥ 3 acute toxicity was also significantly increased, particularly for diarrhea and gastrointestinal toxicity.

Taken together, oral Cape can replace continuous-infusion 5-FU as the standard chemotherapy in preoperative CRT, offering greater convenience without compromising efficacy. In contrast, the routine incorporation of oxaliplatin is not recommended: with the exception of CAO/ARO/AIO-04, major trials have not demonstrated clear clinical benefit, while consistently reporting increased toxicity and reduced treatment compliance.

### 4.6. Interval Between Completion of CRT and Surgery

There is currently no consensus regarding the optimal interval between the completion of preoperative CRT and surgery. Traditionally, surgery has been performed 6–8 weeks after CRT; however, the evidence supporting this timing is limited.

The phase III GRECCAR-6 trial directly compared two waiting intervals—7 weeks versus 11 weeks—to determine their impact on pCR rate [58]. No significant difference in pCR rate was observed between the two arms (15.0% in the 7-week group vs. 17.4% in the 11-week group, *p* = 0.598). However, postoperative complication rates were significantly higher in the 11-week group (44.5% vs. 32%, *p* = 0.040), and the quality of the surgical specimen was poorer after the longer interval (complete mesorectum: 90% vs. 78.7%, *p* = 0.0156). Surgeons also reported greater pelvic fibrosis and mesorectal friability after longer delays, indicating increased technical difficulty. Long-term follow-up showed no interval-related differences in OS, DFS, local recurrence, or distant metastasis [59].

The Stockholm III trial assessed the influence of delaying surgery after SC-RT. Patients were randomized to one of three groups: surgery within one week of RT (SCRT group), surgery delayed for 4–8 weeks after RT (SCRT-delay group), or long-course CRT followed by surgery after 4–8 weeks [60,61,62]. The pCR rate was significantly higher in the SCRT-delay arm (11.8% vs. 1.7%, *p* = 0.001), suggesting that a delayed interval after SC-RT permits additional tumor regression. Local and distant recurrence rates were similar among the three groups; however, acute adverse events and postoperative complications were more common in the SCRT group, implying that a waiting period may mitigate treatment-related morbidity.

A meta-analysis of seven RCTs comprising 3085 patients demonstrated that prolonging the interval increased the pCR rate, with 95% of pCRs achieved within 10 weeks after CRT completion [63]. Using six weeks as a cut-off, the late-surgery interval group (≥6 weeks) exhibited a higher pCR rate than the early-surgery group (<6 weeks) (18.8% vs. 11.6%, *p* < 0.01), without any increase in local recurrence, distant metastasis, DFS, or OS. Patients who achieved pCR had substantially superior 5-year local control (96% vs. 85%), distant metastasis-free survival (89% vs. 67%), DFS (85% vs. 63%), and OS (91% vs. 76%) compared to those who did not achieve pCR.

Another meta-analysis demonstrated that waiting at least eight weeks significantly increased the likelihood of pCR (odds ratio [OR] = 1.41, 95% CI 1.30–1.52, *p* < 0.001) and was associated with more frequent T downstaging (OR = 1.33, *p* = 0.03) and overall downstaging (OR = 1.18, *p* = 0.004) [64]. No differences were observed in R0 resection rates or surgical morbidity (including anastomotic leakage, wound infection, ureteric injury, or venous thromboembolism). Local recurrence rates were comparable, whereas distant metastasis (OR = 0.71, *p* = 0.01) and overall recurrence (OR = 0.76, *p* = 0.04) were significantly lower with the longer interval.

However, several studies have reported that tumors demonstrating poor responsiveness to CRT are associated with worse OS and DFS when the interval to surgery is excessively prolonged [65,66].

In summary, while the optimal interval between CRT and surgery remains under debate, waiting 6–10 weeks appears to improve oncological outcomes by increasing the pCR rate. However, excessive delay may be detrimental for non-responders. In the era of organ-preservation strategies such as watch-and-wait, individualized tailoring of the surgical interval based on tumor biology and response is increasingly important.

## 5. Total Neoadjuvant Therapy (TNT)

For Stage III rectal cancer, the conventional therapeutic strategy has involved administering adjuvant chemotherapy following R0 resection, with the aim of reducing distant recurrence and improving long-term prognosis. However, in patients with advanced rectal cancer accompanied by lymph node metastasis, the initiation of postoperative chemotherapy is frequently challenging, and, when started, completion rates are notably low [36]. Meanwhile, preoperative CRT has been widely implemented as a standard treatment to control local recurrence; however, as previously discussed, CRT alone has a limited impact on long-term outcomes.

In recent years, “total neoadjuvant therapy (TNT)” has attracted attention as a method for overcoming these issues. TNT is a treatment approach in which systemic chemotherapy is integrated preoperatively in addition to preoperative CRT (Table 2). This approach is expected to improve compliance with chemotherapy, facilitate early intervention for micrometastasis, reduce distant recurrence rates, and shorten the interval to the closure of a temporary stoma [67]. Furthermore, in Western practice, the clinical utility of NOM has been widely reported for patients who achieve a cCR after CRT, and the applicability of NOM is expanding with the increased cCR rate due to TNT [68,69,70,71]. Permanent stoma formation, bowel dysfunction, urinary dysfunction, and sexual dysfunction after TME can profoundly impair lifelong quality of life (QOL), even when oncological control is achieved [72,73,74]. The evolution of TNT and NOM is thus shifting the treatment paradigm for LARC from simple recurrence control to the goal of both disease control and QOL maintenance.

Reflecting this trend, the 2024 edition of the NCCN guidelines recommends TNT for patients with Stage II–III rectal cancer [8]. Similarly, the 2018 guidelines of the ESMO recommend the combination of SC-RT and intensified chemotherapy for patients with cT3–T4 disease involving the MRF or with positive lateral pelvic nodes [2].

### 5.1. TNT with Induction Chemotherapy

Within the framework of TNT, the approach of administering systemic chemotherapy prior to preoperative CRT is referred to as “induction chemotherapy” (INCT). This strategy is anticipated to suppress distant recurrence by enabling early intervention for micrometastases.

A pivotal trial demonstrating the efficacy of this approach is the PRODIGE 23 trial [71]. This phase III study enrolled 461 patients aged 18–75 years with cT3 or cT4 rectal cancer located within 15 cm of the anal verge and ECOG performance status 0–1. Patients were randomized into two groups: (i) the TNT group received INCT with six cycles of FOLFIRINOX, followed by preoperative CRT, and then three months of adjuvant chemotherapy with mFOLFOX6 or Cape; (ii) the standard treatment group received preoperative CRT followed by six months of postoperative adjuvant chemotherapy.

The primary endpoint, 3-year DFS, was significantly improved in the TNT group (76% vs. 69%, HR = 0.69, *p* = 0.034), as was 3-year metastasis-free survival (MFS) (79% vs. 72%, HR = 0.64, *p* < 0.02). The pCR rate was also significantly higher in the TNT group (28% vs. 12%, *p* < 0.0001), demonstrating the strong tumor control effect of INCT-TNT. There was no significant difference in the incidence of severe adverse events between the groups (27% vs. 22%, *p* = 0.167), and 92% of patients in the TNT group successfully completed INCT with FOLFIRINOX. Importantly, the TNT group did not experience any disadvantages in terms of surgery rate, perioperative morbidity, or the rate of postoperative adjuvant chemotherapy administration.

Long-term analysis (median follow-up, 82.2 months) further confirmed the superiority of TNT, with the TNT group demonstrating significantly better 7-year DFS (67.6% vs. 62.5%, *p* = 0.048), 7-year MFS (79.2% vs. 72.3%, *p* = 0.021), and 7-year OS (81.9% vs. 76.1%, *p* = 0.033) compared to the standard treatment group [75].

In summary, TNT incorporating INCT with FOLFIRINOX is a promising treatment strategy that offers improved long-term outcomes for patients with cT3 or cT4, M0 LARC, compared to conventional preoperative CRT.

### 5.2. TNT with Consolidation Chemotherapy

Another TNT approach involves the addition of systemic chemotherapy after preoperative CRT and before surgery, known as “consolidation chemotherapy (CNCT)”. This method is expected to maximize tumor downsizing induced by CRT and further improve disease control and pCR rate through CNCT [83].

The Polish II trial was a phase III study that evaluated the efficacy of the CNCT-TNT strategy [70,76]. In this study, 515 patients with fixed cT3 or cT4 rectal cancer were randomized to two groups: the TNT group received SC-RT (25 Gy/5 fractions) followed by CNCT (three cycles of FOLFOX4) and then surgery, while the CRT group received preoperative LC-CRT (50.4 Gy/28 fractions with 5-FU/LV and oxaliplatin). There was no significant difference between groups in the primary endpoints of R0 resection rate (77% vs. 71%, *p* = 0.07) or pCR rate (16% vs. 12%, *p* = 0.17). However, the rate of grade ≥ 3 acute adverse events was significantly lower in the TNT group (75% vs. 83%, *p* = 0.006), indicating better tolerability. In terms of survival, 3-year OS was significantly better in the TNT group (73% vs. 65%, *p* = 0.046), but this difference disappeared at the 8-year follow-up (49% vs. 49%, *p* = 0.38), and there was no difference in DFS (43% vs. 41%, *p* = 0.65).

The RAPIDO trial directly compared the conventional strategy of LC-CRT followed by postoperative adjuvant chemotherapy with TNT consisting of SC-RT plus CNCT in a phase III study [68,77]. Between 2011 and 2016, a total of 920 high-risk rectal cancer patients (cT4, cN2, EMVI-positive, MRF involvement, or lateral pelvic lymph node metastasis) were randomized to either (i) TNT (SC-RT followed by six cycles of CAPOX or nine cycles of FOLFOX4) or (ii) LC-CRT with Cape, followed by optional postoperative CAPOX or FOLFOX4. The primary endpoint, 3-year disease-related treatment failure (DRTF), was significantly lower in the TNT group (23.7% vs. 30.4%, *p* = 0.019). The pCR rate was higher in the TNT group (28% vs. 14%, *p* < 0.0001), and the 3-year distant metastasis rate was lower (20.0% vs. 26.8%, *p* = 0.0048). The completion rate of preoperative chemotherapy in the TNT group was 85%, whereas in the CRT group, the completion rate of postoperative adjuvant chemotherapy was only 37%. The cumulative 3-year local recurrence rate was slightly higher in the TNT group but not significantly different at this time (8.3% vs. 6%, *p* = 0.12). Importantly, these improvements were achieved without negatively affecting 3-year QOL, bowel function, or late adverse events. The RAPIDO trial demonstrated the usefulness of TNT (SC-RT plus CNCT) as a breakthrough in treating high-risk rectal cancer.

However, extended follow-up of the RAPIDO trial (median 5.6 years) revealed a significantly higher local recurrence rate among R0/R1 resection cases in the TNT group (10.2% vs. 6.1%, *p* = 0.027) [78]. The rate of mesorectal breach was also higher in the TNT group (11% vs. 6%, *p* = 0.022), suggesting that the prolonged interval between SC-RT and surgery may result in tissue fragility or fibrosis, compromising specimen quality and increasing the risk of local recurrence, despite improved systemic control.

The STELLAR trial also compared CNCT-TNT and LC-CRT in a phase III RCT [69]. From 2015 to 2018, 599 patients with cT3/cT4 or node-positive rectal cancer were randomized to either TNT (SC-RT plus four cycles of CAPOX) or CRT (LC-CRT with Cape), with TME performed 6–8 weeks after neoadjuvant therapy and postoperative adjuvant chemotherapy given in both groups. TNT was non-inferior for 3-year DFS (64.5% vs. 62.3%, *p* < 0.001 for non-inferiority) and superior for 3-year OS (86.5% vs. 75.1%, *p* = 0.033). MFS and local recurrence rates were comparable. TNT produced a higher pCR rate (21.8% vs. 12.3%, *p* = 0.002) without increasing postoperative morbidity (14.0% vs. 15.7%, *p* = 0.625), although grade ≥ 3 acute toxicity was more common (26.5% vs. 12.6%, *p* < 0.001).

Taken together, CNCT-TNT offers advantages such as improved tumor shrinkage, increased pCR rates, and potentially better survival compared to conventional LC-CRT plus postoperative chemotherapy. However, given the potential risks of impaired local control and increased acute toxicity, careful risk assessment and individualized application are warranted.

### 5.3. Comparison of TNT Treatment Sequences: INCT vs. CNCT

TNT is increasingly accepted as a strategy that can improve long-term survival in LARC, which was previously difficult to achieve with conventional preoperative RT/CRT alone. However, the optimal sequencing of systemic chemotherapy within TNT—whether INCT (systemic chemotherapy prior to CRT) or CNCT (chemotherapy following CRT)—remains a subject of ongoing debate.

The CAO/ARO/AIO-12 trial was a prospective, randomized phase II study that directly compared these two TNT sequences [79,80]. Among 311 patients with cT3/cT4 and/or node-positive LARC, patients were randomized to either the INCT group (INCT → CRT → surgery) or the CNCT group (CRT → CNCT → surgery). In both groups, TME was performed as radical surgery; a watch-and-wait strategy was not generally permitted, and postoperative adjuvant chemotherapy was not recommended. The primary endpoint, which was the pCR rate, was 17% (95% CI: 12–24%) in the INCT group and 25% (95% CI: 18–32%) in the CNCT group, with only the CNCT group meeting the predefined threshold of ≥15% (*p* < 0.001). CRT-related grade 3–4 adverse events were significantly higher in the INCT group (37% vs. 27%), and CRT completion was higher in the CNCT group. Chemotherapy-related grade 3–4 adverse events were similar between groups (22% vs. 22%), but the completion rate of chemotherapy was higher in the INCT group (92% vs. 85%). In the 3-year follow-up, there were no significant differences between groups in DFS (73% vs. 73%, *p* = 0.82), local recurrence (6% vs. 5%, *p* = 0.67), or distant metastasis (18% vs. 16%, *p* = 0.52). These results indicate that CNCT achieves a higher pCR rate without compromising long-term outcomes, suggesting its potential as a preferred TNT sequence.

Furthermore, the OPRA trial was the first prospective phase II trial to investigate treatment optimization based on a watch-and-wait strategy (NOM) after TNT [81,82]. In this study, 324 rectal cancer patients were randomized to either the INCT or CNCT groups. After TNT, patients who achieved a cCR or near-cCR were offered NOM. The primary endpoint, which was 5-year DFS, did not differ significantly between the groups (71% vs. 69%, *p* = 0.68). However, 5-year TME-free survival was significantly higher in the CNCT group (54% vs. 39%, *p* = 0.012), and the local regrowth rate among NOM patients was lower in the CNCT group (29% vs. 44%, *p* = 0.02). Notably, among patients who underwent salvage TME for local regrowth after NOM and those who underwent TME for incomplete CR after TNT, 5-year DFS was comparable (64% vs. 64%, *p* = 0.94).

Collectively, these findings suggest that for patients aiming for NOM after TNT, the CNCT approach may provide greater opportunities for organ preservation. Overall, CNCT-TNT appears to yield higher pCR/cCR rates and aligns well with NOM strategies, although careful individual risk–benefit assessment remains essential.

### 5.4. Comparative Analysis and Interpretation of Recent Clinical Trials on TNT

The sequencing debate between INCT and CNCT within TNT remains pivotal in the treatment of LARC. Although no direct large-scale phase III trials have conclusively compared these sequences, recent clinical studies provide valuable insights regarding their relative efficacy, patient selection, and clinical significance.

INCT, exemplified by the PRODIGE 23 trial, employs systemic chemotherapy upfront, primarily targeting micrometastases early in the treatment course [71,75]. PRODIGE 23 demonstrated a significant improvement in distant control and long-term survival metrics, notably DFS and OS, emphasizing the benefits of early systemic intervention for patients with advanced-stage tumors or significant nodal involvement. Thus, induction TNT can be particularly advantageous in populations with high-risk micrometastatic profiles.

Conversely, CNCT, as evaluated in the RAPIDO and STELLAR trials, emphasizes enhanced local tumor response and higher pCR rates by administering chemotherapy during the interval between the completion of CRT and surgery, thereby intentionally prolonging the waiting period after CRT before surgery [68,69,77]. These studies collectively indicate that CNCT is beneficial in maximizing tumor regression, which is particularly useful in patients with bulky local disease or when aiming for organ preservation strategies.

Clinically significant distinctions between INCT and CNCT emerged more explicitly from the randomized phase II CAO/ARO/AIO-12 trial, which directly compared these sequences [79,80]. This study confirmed higher pCR rates in the CNCT arm; however, this improvement did not translate into better DFS or OS. These findings indicate that although CNCT may enhance immediate tumor downsizing, the sequencing effect alone may have a minimal impact on long-term survival outcomes. Notably, this trial also revealed distinct differences in treatment compliance and toxicity profiles depending on the sequencing; specifically, chemotherapy compliance rates were higher when administered first (INCT), while CRT tolerability improved with upfront CRT (CNCT) [80]. Consequently, treatment sequence can meaningfully influence therapy adherence and patient tolerance.

The OPRA trial further illuminated the clinical implications of sequence selection, demonstrating that CNCT-TNT significantly improved organ preservation rates compared to INCT-TNT [81,82]. This pivotal finding substantially advanced the adoption of NOM strategies. By prioritizing CNCT, clinicians significantly increased the probability of achieving and maintaining a cCR, fundamentally shifting the therapeutic paradigm toward organ preservation. Although local tumor regrowth remains a consideration in NOM, OPRA importantly verified that salvage surgery after local regrowth offers oncological safety comparable to immediate radical surgery.

In summary, optimal sequencing within TNT should align with individualized treatment goals and tumor biology. INCT provides pronounced systemic control advantages suitable for micrometastatic disease, while CNCT maximizes local tumor response and is integral for NOM strategies. A clear understanding of these differential effects, including compliance rates, adverse event profiles, and tumor regression dynamics, is critical for personalized patient selection and optimized clinical outcomes. Future randomized trials directly comparing these sequences will further refine strategic decisions, enhancing both oncologic and patient-centered outcomes in LARC treatment.

### 5.5. Molecular-Targeted Agents Within TNT

In metastatic CRC, molecular-targeted agents, such as inhibitors of vascular endothelial growth factor (VEGF) and monoclonal antibodies against the epidermal growth factor receptor (EGFR), have led to improved survival. By contrast, in the adjuvant setting, these agents have not demonstrated significant long-term benefits, and current guidelines do not recommend their incorporation into standard preoperative therapy for rectal cancer [8]. Nevertheless, several phase II studies incorporating molecular targeted therapies into TNT have suggested promising tumor control effects.

The AVACROSS trial was a multicenter phase II study that evaluated the efficacy and safety of TNT combined with bevacizumab in patients with LARC who had a poor prognosis, as determined by MRI [84]. Forty-seven patients received INCT with four cycles of XELOX plus bevacizumab, followed by bevacizumab-containing CRT, and TME was performed 6–8 weeks later. The pCR rate was 36%, and the R0 resection rate was 98%, indicating favorable outcomes. However, postoperative complications occurred in 58% of cases, and 24% required reoperation, mainly due to anastomotic leakage.

The INOVA trial was a phase II randomized trial that similarly assessed TNT with bevacizumab [85,86]. Ninety-one patients were allocated to the following arms: (i) induction FOLFOX4 plus bevacizumab followed by bevacizumab-CRT (Arm A) or (ii) bevacizumab-CRT alone (Arm B). Arm A achieved a higher pCR rate (23.8% vs. 11.4%), exceeding the predefined threshold of ≥10% (*p* = 0.015). Five-year DFS and OS favored Arm A, but post-operative fistula—again mainly anastomotic leaks—occurred in 20% of both arms, underscoring surgical risk.

The TRUST study incorporated FOLFOXIRI plus bevacizumab as INCT, followed by bevacizumab-containing CRT [87]. Among 48 patients, the pCR rate was 36.4%, and the overall response rate was 88.9%, with a 2-year DFS of 80.5%. However, high rates of grade 3–4 neutropenia (41.6%) and diarrhea (12.5%) were observed, and postoperative morbidity remained considerable (anastomotic leakage 18%, any complication 32%).

These results suggest that VEGF inhibitors in TNT may offer substantial oncological benefits but carry significant trade-offs in the risk of postoperative complications. Careful patient selection and intensive perioperative management are essential, and high-quality phase III trials are necessary before anti-angiogenic agents can be routinely incorporated into TNT protocols.

The utility of anti-EGFR antibodies has been evaluated in the EXPERT-C trial [88]. This randomized phase II trial enrolled 165 patients with resectable rectal cancer of poor prognosis as determined by MRI, comparing CAPOX plus CRT plus postoperative CAPOX with the same regimen with the addition of cetuximab (CAPOX + C group). In patients with KRAS/BRAF wild-type tumors, the primary endpoint, which was the pCR rates, was similar (9% vs. 11%, *p* = 1.0), but the response rates at the end of preoperative chemotherapy (51% vs. 71%, *p* = 0.038) and after CRT (75% vs. 93%, *p* = 0.028) were significantly higher in the CAPOX + C group. OS was also significantly longer in the CAPOX + C group (HR = 0.27, *p* = 0.034), although progression-free survival was not. In the overall cohort, no significant differences were observed in primary or secondary endpoints. Although skin toxicity and diarrhea were increased in the CAPOX + C group, serious adverse events did not frequently lead to discontinuation, and the completion rate for adjuvant CAPOX was 65%. While this study suggests a certain utility for anti-EGFR antibodies, since the primary endpoint was not met, routine inclusion in TNT is not currently recommended.

In summary, early-phase studies suggest that the addition of molecular-targeted agents to TNT can enhance tumor regression; however, this apparent benefit is offset by increased acute toxicity and a higher incidence of postoperative complications, particularly anastomotic leakage associated with VEGF inhibitor regimens. Notably, the clinical efficacy of molecular-targeted agents in this setting appears to be lower than theoretically anticipated. This discrepancy may be largely attributable to the current limitations in the predictive accuracy of available biomarkers. For example, no established biomarkers exist for bevacizumab, resulting in the absence of clear patient selection criteria for its use. Addressing these challenges will require further translational research and a deeper elucidation of the molecular profiles of rectal cancer. These issues, including the need for improved patient stratification and biomarker-driven approaches, will be discussed in Section 9, “Unresolved Issues in Preoperative Treatment for LARC”.

## 6. NAC: Omission of RT

To date, the strategies for preoperative therapy for LARC have largely followed a stepwise escalation paradigm that intensifies treatment in pursuit of improved local control and long-term survival. Preoperative CRT, the cornerstone of these approaches, has proven effective in increasing resectability and reducing local recurrence but is also associated with cumulative long-term adverse events [89,90,91]. Pelvic irradiation can induce fibrosis and tissue degeneration that not only complicate surgical dissection but may also weaken sphincter function and impair bowel function. Moreover, late toxicities, such as enteropathy, sexual dysfunction, pathological fractures, and secondary malignancies, must not be overlooked. Additionally, limitations in imaging and clinical staging may result in CRT being applied indiscriminately, even in cases at low risk of recurrence, raising concerns about overtreatment. In this context, NAC alone—selectively omitting radiation—has garnered increasing interest as a personalized approach (Table 3). With appropriate patient selection, NAC offers the clear advantage of avoiding radiation-related toxicity and, as systemic chemotherapy, provides early intervention for micrometastases and circulating tumor cells, as well as improved compliance.

The PROSPECT trial was a phase II/III, randomized, multicenter study that investigated the non-inferiority of NAC (upfront FOLFOX) to standard preoperative CRT in 1194 patients with T2N+, T3N0, or T3N+ tumors deemed eligible for sphincter-preserving surgery [92]. In the NAC arm, CRT was reserved only for cases with <20% tumor shrinkage or intolerable toxicity. The primary endpoint, 5-year DFS, was 80.8% in the NAC arm and 78.6% with CRT, meeting the non-inferiority criterion (HR = 0.92; 90.2% CI 0.74–1.14; non-inferiority *p* = 0.005). Local recurrence rates were extremely low in both groups (1.8% vs. 1.6%), with no significant differences in OS, R0 resection rate, or pCR rate. Notably, 89.6% of patients in the NAC group ultimately avoided CRT.

The FOWARC trial, a phase III trial conducted in China, enrolled 495 patients with Stage II/III rectal cancer and randomized them to receive (i) 5-FU plus RT, (ii) mFOLFOX6 plus RT, or (iii) mFOLFOX6 alone [93,94]. The primary endpoint, 3-year DFS, was 72.9%, 77.2%, and 73.5% for the three groups, respectively, with no significant differences (*p* = 0.709). Although the pCR rate was lowest with mFOLFOX6 alone (6.6% vs. 27.5% with mFOLFOX6 + RT), down-staging to ypStage 0–1 occurred in 35.5%, which was comparable to the 5-FU + RT arm (37.1%). At the 10-year follow-up, no significant differences were observed in DFS (52.5%, 62.6%, 60.5%; *p* = 0.56), local recurrence (10.8%, 8.0%, 9.6%; *p* = 0.57), or OS (65.9%, 72.3%, 73.4%; *p* = 0.90) among the three groups [95]. Patients who achieved pCR had excellent outcomes, with a 10-year DFS of 84.3%, local recurrence of 3.0%, and OS of 92.4%. The mFOLFOX6-alone group experienced fewer adverse events, lower rates of anastomotic leakage and defecation disorders, and significantly better QOL indices, including Wexner score, stool frequency, and nocturnal incontinence.

The CONVERT trial was a phase III non-inferiority RCT that compared NAC with CAPOX versus preoperative LC-CRT in patients with LARC without MRF involvement [96]. Among 663 enrolled patients, no significant differences were found in the pCR rate (11.0% vs. 13.8%, *p* = 0.333), ypStage 0–1 rate (40.8% vs. 45.6%, *p* = 0.27), R0 resection rate, or sphincter preservation rate. The NAC group had lower rates of perioperative distant metastasis (0.7% vs. 3.1%, *p* = 0.03) and prophylactic stoma creation (52.2% vs. 63.6%, *p* = 0.008). Long-term outcomes are awaited.

These findings, in the context of advances in high-resolution MRI staging, standardized TME, and improved systemic chemotherapy, suggest that it may be time to reconsider the routine use of CRT for all cases of LARC. NAC alone may be a reasonable and safe treatment option, especially for intermediate-risk patients for whom preservation of anal function and QOL is relatively important. Conversely, caution is warranted in high-risk cases, such as those with T4 tumors or very low-lying lesions, where the benefits of radiation may still outweigh its risks.

## 7. Non-Operative Management (NOM)

With the widespread adoption of preoperative CRT and TNT, an increasing proportion of rectal cancer patients are now achieving pCR. Traditionally, even patients with pCR have undergone standard TME; however, rectal resection is associated with postoperative complications and functional impairments that can significantly impact patients’ QOL. Anal dysfunction is almost inevitable, and urinary and sexual dysfunctions are also frequently observed [72,73,74]. Severe postoperative complications, such as anastomotic leakage, can prolong hospitalization and result in poor prognosis, while the creation of a permanent or temporary stoma can greatly diminish QOL. Conversely, patients who have achieved pCR have demonstrated excellent long-term outcomes [97], leading to the proposal and increasing adoption of a “NOM” strategy—particularly in Western countries. In this approach, patients who achieve cCR are closely observed without undergoing immediate surgery.

### 7.1. Diagnosis of cCR and Selection of Candidates for NOM

The accurate identification of a cCR is crucial for the safe implementation of NOM. Three modalities are essential for diagnosing cCR: digital rectal examination (DRE), endoscopy, and MRI. Diagnostic accuracy is enhanced by a combined comprehensive assessment, rather than reliance on any single modality [98]. On DRE, there should be no palpable mass or mucosal irregularity. Endoscopically, findings consistent with cCR include a flat, white scar with telangiectasia and the complete resolution of any ulcer or mass. Notably, a negative mucosal biopsy cannot exclude residual cancer in deeper layers, while a positive biopsy may occasionally resolve spontaneously; therefore, the value of biopsy as a diagnostic adjunct is limited. MRI, particularly with T2-weighted and diffusion-weighted imaging (DWI), plays a pivotal role. On T2-weighted images, the assessment of the relative proportion of tumor signal (isodensity) and fibrotic scar (low density) is used to evaluate the MRI-based tumor regression grade (mrTRG). cCR is characterized by the complete disappearance of the tumor signal, leaving only fibrosis [99]. On DWI, disappearance of the high signal corresponding to the tumor further supports a cCR diagnosis [100].

Currently, there are no universally accepted, standardized criteria for cCR diagnosis; however, diagnostic algorithms proposed by Habr-Gama et al. and the Memorial Sloan Kettering Cancer Center (MSKCC) are widely referenced in clinical practice [101,102]. These definitions, which integrate findings from DRE, endoscopy, and MRI, are commonly used when considering a watch-and-wait strategy.

Selection for NOM should be determined carefully by an experienced multidisciplinary team, including colorectal surgeons, radiation oncologists, medical oncologists, gastroenterologists, and radiologists. It is essential to incorporate patient preferences, potential impact on QOL, and the feasibility of a robust follow-up system as integral components of the decision-making process, especially when NOM is being introduced.

### 7.2. Outcomes of NOM

The clinical efficacy of NOM was first reported by Habr-Gama et al. in Brazil [103]. Among 265 patients with lower rectal cancer who received preoperative CRT between 1991 and 2002, 71 (26.8%) achieved a cCR at eight weeks and were managed with NOM. The remaining 194 patients with an incomplete clinical response (iCR) underwent TME, of whom 22 (8.3%) proved to have a pCR. The five-year OS and DFS in the cCR group were 100% and 92%, respectively, which were comparable to or better than those in the pCR group (88% and 83%). Only two patients in the cCR group developed local regrowth, both of which were successfully managed with local resection or brachytherapy, with no subsequent recurrence.

The UK OnCoRe (Oncological Outcomes after Clinical Complete Response in Patients with Rectal Cancer) project prospectively compared NOM with surgery using propensity-score matching across multiple centers [104]. Among 333 patients registered from 2005 to 2015, 129 were managed with NOM after cCR and 228 underwent TME. The cumulative 3-year local regrowth rate in the NOM group was 38% (95% CI, 30–48); however, 88% of regrowths were amenable to salvage resection, resulting in an R0 margin in 77% of cases. The primary endpoint, which was non-regrowth DFS, was 88% at three years in the NOM group, demonstrating non-inferiority compared to the TME group (78%, *p* = 0.043). Three-year OS was likewise non-inferior (96% vs. 87%; *p* = 0.024), and the stoma-free survival rate was significantly higher with NOM (74% vs. 47%; *p* < 0.0001).

Additionally, the International Watch & Wait Database (IWWD), which registers patients from 47 centers in 15 countries, has reported outcomes of NOM [105]. Among 880 cCR patients enrolled between 2015 and 2017, the 2-year cumulative incidence of local regrowth was 25.2% (95% CI, 22.2–28.5). Of these regrowths, 88.3% were diagnosed within the first two years, and 96.7% remained confined to the bowel wall. Distant metastases occurred in 71 patients (8%). The five-year OS and disease-specific survival rates were 84.6% (95% CI, 80.9–87.7) and 93.8% (95% CI, 90.9–95.9), respectively. Salvage therapy consisted of TME in 77.7% and local excision in 22.3%, yielding an overall R0 rate of 87.8%.

A report by Nasir et al. summarized the outcomes of salvage surgery for local regrowth, finding that minimally invasive techniques (laparoscopic or robotic) were feasible for all cases. Short-term surgical outcomes, pathology, and three-year oncological outcomes were comparable to those who underwent immediate surgery after CRT [106].

On the other hand, several reports indicate a higher rate of distant metastasis in cases with local regrowth [107,108]. An analysis of the IWWD involving 793 cCR cases showed that 24.1% of patients who developed local regrowth also experienced distant metastasis, indicating that local regrowth is an independent risk factor for distant metastasis (HR = 4.8, 95% CI: 3.14–7.49). These findings underscore the necessity for stringent surveillance, particularly during the early period when regrowth is most likely to occur.

The recent prospective phase II CAO/ARO/AIO-16 trial reported the outcomes of NOM after TNT in patients achieving cCR [109]. Ninety-one patients with cT1-2N1-2 or cT3a-dN0/N1-2 disease received CNCT-TNT, with response evaluation on day 106. Forty percent (36 patients) met NOM criteria; 44% later developed local regrowth, but all were salvaged with R0 TME, and non-regrowth DFS was equivalent to that in the upfront-surgery cohort. Bowel function scores (LARS and Wexner) were significantly better in patients who avoided TME.

Collectively, these data demonstrate that in carefully selected patients who attain cCR, NOM yields long-term oncologic outcomes comparable to TME while sparing most patients the morbidity of major pelvic surgery, permanent or temporary stomas, and associated decrements in QOL. Although local regrowth occurs in roughly one in four patients, timely detection and effective salvage surgery can preserve survival outcomes. Therefore, NOM should not be viewed simply as the avoidance of surgery but as a strategy that must be coupled with rigorous, protocol-driven surveillance to ensure patient safety.

### 7.3. QOL in NOM

While the NOM strategy increases the opportunity to avoid surgery in rectal cancer, its long-term impact on patient-reported QOL remains a critical component of clinical decision-making. A prospective cohort study conducted by a Dutch group provides valuable real-world data on QOL outcomes following NOM [110]. In this study, 278 patients from 13 centers in the Netherlands and Belgium were followed for at least 24 months, of whom 221 (approximately 80%) were managed with NOM. Overall QOL scores were favorable, with significant improvements in social and emotional functioning observed at 24 months post-treatment. However, some patients continued to experience bowel dysfunction: at 24 months, approximately 25% reported severe bowel dysfunction, and the prevalence of severe fecal incontinence increased from 10.9% at 3 months to 22.2% at 24 months. Additionally, about 30% of male patients reported severe erectile dysfunction, highlighting the persistent impact on sexual function.

The impact of additional treatments has also been reported. Among patients who underwent local excision (6.5%), overall QOL scores were similar to those managed with NOM alone; however, the frequency of major low anterior resection syndrome (LARS) was high (55.6%), indicating substantial bowel dysfunction. In patients who subsequently underwent TME (14%), scores for physical, social, and emotional function were all significantly lower than those in the NOM group, accompanied by increased pain, fatigue, urinary dysfunction, appetite loss, and financial difficulties—reflecting a multifaceted decline in QOL.

These data underscore the importance of balanced and thorough counseling when offering NOM. While NOM can preserve or even enhance certain domains of QOL, clinicians must also inform patients about the risks of bowel and sexual dysfunction and the potential need for salvage therapy in the event of local regrowth. Shared decision-making should carefully integrate these long-term QOL considerations alongside oncological safety.

## 8. Immune Checkpoint Inhibitors for LARC

A cornerstone of personalized optimization in LARC treatment is precision medicine guided by molecular characteristics. Notably, patients with deficient MMR (dMMR) or MSI-high tumors have shown remarkable responsiveness to ICIs, raising the possibility of omitting surgery and CRT altogether. In a phase II trial by Cercek et al., all 12 patients with dMMR LARC who completed six months of anti-PD-1 monotherapy (dostarlimab) achieved cCR without requiring surgery or CRT, with no recurrences or progression at a median 12-month follow-up [111]. Treatment-related adverse events were mild, and no grade ≥ 3 events occurred. Similarly, a subsequent multicenter phase II study in Stage I–III dMMR solid tumors (including 49 LARC cases) reported that neoadjuvant dostarlimab enabled NOM: all rectal cancer patients achieved cCR, and ~76% sustained cCR for ≥1 year [112]. None experienced unresectable progression, preserving the “window of opportunity” for curative resection, and grade ≥ 3 adverse events were rare. These findings highlight the potential for immunotherapy to enable organ preservation in rectal cancer; however, this dramatic benefit has been confined to the MSI-high/dMMR population thus far, which comprises only a small fraction (roughly 5%) of rectal cancers [20,22]. Such tumors are characterized by a high mutation burden and dense immune infiltration, correlating with their exceptional responsiveness to ICIs [22].

In contrast, the majority of LARC cases are proficient MMR/microsatellite-stable (pMMR/MSS) and lack these immunogenic features, and single-agent immunotherapy has shown minimal activity in this group [21,25]. Notably, a randomized phase II trial (NRG-GI002) that added anti-PD-1 antibody pembrolizumab to TNT in an unselected LARC population did not significantly improve outcomes, failing to meet its primary endpoint for tumor response [113]. Similarly, early endpoints like the neoadjuvant rectal (NAR) score and pCR rate were not significantly improved with PD-1 blockade in that study, and further evaluation of upfront immunotherapy in unselected LARC was not recommended [113]. These results underscore the lack of established patient selection criteria beyond MSI-high/dMMR status, as well as the need for more effective biomarkers to identify which patients with rectal tumors might benefit from immune-based therapy.

Current predictive biomarkers for immunotherapy in rectal cancer have important limitations. MSI-high/dMMR is the clearest positive predictor, but it is rare in LARC, and not all MSI-high/dMMR tumors achieve complete remission [25]. Other markers, such as programmed death-ligand 1 (PD-L1) expression and tumor mutational burden (TMB), have been explored but have not demonstrated robust utility in CRC. Unlike in other solid tumors, PD-L1 immunohistochemical status has not reliably predicted a response to PD-1 therapy in colorectal cancer [25]. TMB is generally elevated in MSI-high/dMMR tumors (and in occasional ultra-mutated cases like POLE-mutant CRC), yet most MSS rectal cancers have low TMB and remain unresponsive to ICIs [21,22]. Thus, no alternative immunotherapy biomarkers are validated for routine use in MSS rectal cancer at this time. This leaves a significant gap in our ability to tailor immunotherapy to the right patients: beyond the MSI-high/dMMR subset, there is a lack of consensus on which, if any, LARC patients should receive ICIs in the neoadjuvant setting. Identifying responsive subpopulations within MSS tumors is an urgent research priority. Molecular subtyping provides some insights—for instance, the CMS1 (which includes MSI-high tumors) features an inflamed, immune-rich microenvironment, whereas the prevalent CMS2/CMS3 subtypes of rectal cancer are immune cold and more likely to resist immunotherapy [21,22]. However, such classifications have yet to translate into actionable selection criteria for immunotherapy in LARC.

Another challenge is assessing treatment response and outcomes unique to neoadjuvant immunotherapy. In cases where immunotherapy renders the tumor clinically undetectable and surgery is omitted, confirming a pCR is difficult. Standard cCR criteria (based on endoscopy, MRI, and biopsies) have inherent limitations in sensitivity for microscopic residual disease. In the MSI-high/dMMR rectal cancer trials, patients proceeded without resection following clinical remission, necessitating rigorous endoscopic/imaging surveillance to detect any regrowth [112]. The omission of surgery also means that traditional histopathological grading of tumor regression cannot be performed, complicating the evaluation of response depth.

Furthermore, the toxicity profile of ICIs must be carefully considered in the curative LARC context. Anti-PD-1 and anti-CTLA-4 agents can cause immune-related adverse events (irAEs) that affect the colon, endocrine organs, skin, liver, and other systems. In the pembrolizumab–chemoradiotherapy trial, 43% of patients experienced irAEs (e.g., thyroiditis, dermatitis, hepatitis, or colitis), although only ~4% were high-grade (no grade 4–5 events) [113]. This illustrates that combining immunotherapy with intensive local therapy is feasible but not without risk. Any expansion of ICIs to a broader rectal cancer population would require the vigilant management of irAEs and careful patient counseling. On the other hand, initial trials in dMMR patients have reported remarkably mild toxicity profiles, suggesting that in biomarker-selected cases the risk–benefit balance of neoadjuvant immunotherapy can be favorable [111,112]. Overall, the safety considerations reinforce the importance of prudent patient selection to avoid exposing those unlikely to benefit to potential harm.

## 9. Unresolved Issues in Preoperative Treatment for LARC

Preoperative therapy for LARC has become increasingly diverse, now encompassing RT alone, CRT, NAC, TNT, and even NOM, marking a new era of individualized treatment selection. Nevertheless, several critical questions remain unanswered, and resolving them is essential for future therapeutic refinement.

### 9.1. Selection of Optimal Regimens for Total Neoadjuvant Therapy

Within the context of TNT, several unresolved questions remain, including the optimal chemotherapy regimen—specifically, whether doublet or triplet therapy is superior—as well as the ideal combination and sequencing of RT modalities. Ongoing clinical trials are expected to provide critical insights into these issues. For example, the Janus Rectal Cancer trial (NCT05610163), a phase II/III randomized study, is comparing two TNT regimens following long-course CRT: doublet chemotherapy (mFOLFOX or CAPOX) versus triplet chemotherapy (mFOLFIRINOX), with the cCR rate (phase II) and DFS (phase III) as the primary endpoints [114]. Similarly, the ENSEMBLE trial (NCT05646511 and jRCTs031220342) is a phase III study comparing doublet (CAPOX) versus triplet (CAPOXIRI) CNCT following SC-RT, with organ preservation-adapted DFS as the primary endpoint [115].

In addition to the choice of chemotherapy regimen, it remains unclear whether SC-RT-based TNT confers a higher risk of local recurrence compared to LC-CRT-based TNT. While the RAPIDO trial reported a significantly higher local recurrence rate after SC-RT + CNCT, the Polish II and STELLAR trials found no difference between SC-RT and LC-CRT at the same CNCT-TNT setting [69,70,78]. To directly address this question, the ongoing ACO/ARO/AIO-18.1 trial (NCT04246684) is comparing LC-CRT + CNCT with SC-RT + CNCT based on the hypothesis that the former may provide superior 3-year organ preservation rates.

As results from these ongoing studies become available, the further optimization and personalization of preoperative treatment for LARC can be anticipated, ultimately leading to improved efficacy and safety profiles.

### 9.2. Challenges Associated with cCR Diagnosis and Surveillance Strategies in NOM

Despite growing enthusiasm for NOM in rectal cancer, several unresolved issues continue to limit its broader adoption.

First, there is a lack of standardization in defining a cCR or “near-cCR” across studies, which complicates patient selection and outcome comparisons. Recent international consensus efforts have begun addressing this variability—for example, a 2024 Delphi panel of experts reached agreement on uniform cCR/near-cCR criteria, endorsing a three-tiered near-cCR classification based on the likelihood of achieving eventual complete response and clarifying that “near-cCR” is only a transient designation within the first 6 months post-therapy [116].

Second, accurately restaging tumors after neoadjuvant treatment remains challenging due to the diagnostic limitations of available tools (MRI, DRE, and endoscopy) in the absence of reliable biomarkers. Restaging MRI frequently demonstrates only fair concordance with final pathology. Gefen et al. found fair agreement for T stage status (kappa = −0.316) and only slight concordance for N stage (kappa = −0.11) and CRM status (kappa = 0.089) when compared with pathological assessment. Notably, 73% of patients with pathologically node-positive cases were misclassified as node-negative on restaging imaging [117]. DRE and endoscopy can likewise be confounded by post-treatment fibrosis or subtle residual mucosal abnormalities, leading to both false negatives and false positives in determining cCR. Given these shortcomings, emerging technologies are being explored to improve response assessment. Radiomics, for instance, uses quantitative image features from pre- and post-therapy MRI to predict pathological response. Early studies and meta-analyses have suggested that MRI-based radiomics can achieve encouraging accuracy (AUC = 0.91) for pCR prediction, but such models remain unvalidated and are not yet part of routine care [118]. Similarly, circulating tumor DNA (ctDNA) has shown promise as a minimally invasive biomarker of residual disease: post-neoadjuvant therapy, ctDNA is detectable in only ~15–20% of patients (versus ~75% at baseline), and its persistence correlates with higher risk of tumor regrowth and lower progression-free survival rates [119]. However, the clinical utility of ctDNA for guiding NOM is still unproven, and no uniform molecular marker is currently available to definitively confirm a cCR.

Third, there is considerable heterogeneity in surveillance strategies after NOM, and real-world compliance with intensive follow-up protocols is poor. In the absence of standardized guidelines, follow-up schedules vary widely between institutions [103,104]. Notably, a recent cohort study found that fewer than half of patients adhered to all recommended surveillance visits in the first year of a watch-and-wait program [120]. This lack of uniformity and suboptimal compliance in post-NOM surveillance not only risks delayed detection of local regrowth but also complicates the generalizability of outcomes reported by expert centers. To address these challenges, an international expert panel was convened in 2021 to establish the first consensus recommendations on follow-up protocols for rectal cancer organ preservation strategies [121]. This panel emphasized the urgent need for standardized definitions and schedules, recommending regular assessments—including DRE, endoscopy, pelvic MRI, and chest/abdominal CT—particularly within the first three years when local regrowth risk is highest. The implementation of these consensus protocols is expected to improve the consistency and quality of outcome reporting across centers.

### 9.3. The Need for Improved Patient Stratification and Biomarker-Driven Approaches

Despite significant advances in the neoadjuvant treatment of LARC, the integration of molecularly targeted agents and ICIs into standard regimens remains limited by substantial translational challenges. A major hurdle is the lack of robust, clinically validated biomarkers for optimal patient selection, which hampers the precise application of these therapies.

Recent large-scale genomic and transcriptomic studies have elucidated the profound molecular heterogeneity of CRC, culminating in the establishment of CMS and subsequent refinements, such as the iCMS and PDS. These molecular classifications have provided crucial insights into tumor biology, revealing subpopulations with distinct prognoses and therapeutic vulnerabilities [20,21,22,23,24,122]. For example, MSI-high/dMMR tumors, which account for approximately 10–15% of all colorectal cancers (and only ~5% of rectal cancers), demonstrate exceptional sensitivity to immune checkpoint blockade, leading to dramatic and durable responses in both advanced and, more recently, early-stage disease [21,25,122]. However, reliable predictive biomarkers for the vast majority of patients with MSS tumors—who represent the bulk of rectal cancer cases—remain elusive.

While the use of EGFR inhibitors is guided by well-established negative predictive markers (KRAS, NRAS, and BRAF wild-type status), there are currently no robust predictive biomarkers for response to anti-angiogenic agents such as bevacizumab, and positive selection criteria for these therapies remain undefined [21]. This has resulted in largely empirical treatment decisions, contributing to variable clinical benefit and unnecessary toxicity. ICIs, while highly effective in MSI-high/dMMR populations, have shown minimal efficacy in MSS tumors, underlining the need for new biomarkers capable of identifying immunotherapy-responsive subgroups among these patients [25].

The emergence of ctDNA-based molecular residual disease (MRD) detection as a prognostic and potentially predictive biomarker marks a significant advance in real-time risk stratification, enabling the identification of patients at high risk of recurrence after neoadjuvant or definitive therapy [123,124]. In the CIRCULATE-Japan GALAXY study, ctDNA positivity during the surveillance period was the strongest predictor of poor DFS and OS, regardless of traditional clinicopathologic or actionable biomarker status, suggesting its utility in guiding treatment intensification or de-escalation strategies [123].

Recent multiomics approaches—integrating genomic, transcriptomic, and immunological profiling—are accelerating the transition from experience-based to precision medicine in CRC. Projects such as SCRUM-Japan GI-SCREEN and MONSTAR-SCREEN have demonstrated the feasibility of nationwide biomarker screening, facilitating patient allocation to genotype-matched clinical trials and contributing to drug development and approval in advanced solid tumors [124].

Looking forward, comprehensive molecular and immunoprofiling, including single-cell sequencing and integrated multiomics, are expected to further refine our ability to stratify patients and tailor therapy. Novel subtypes such as iCMS and PDS, which capture the epithelial and pathway-level diversity of CRC, offer a promising framework for more nuanced risk assessment and therapeutic targeting [23,24]. Furthermore, the combination of dynamic biomarkers such as ctDNA MRD with static molecular profiles may enhance both prognostic and predictive accuracy.

To enable a paradigm shift from empirical management to personalized treatment, it is essential to establish the clinical utility and validity of these emerging biomarkers in prospective trials, ensure accessibility to advanced profiling technologies, and develop standardized criteria for their interpretation and integration into clinical workflows. Ultimately, such advances hold the potential to optimize outcomes by selectively intensifying or de-escalating therapy, reducing overtreatment, and preserving organ function in patients with rectal cancer.

## 10. Conclusions

The management of LARC stands at a pivotal crossroads, propelled by major advances in multimodal treatment, evolving surgical paradigms, and breakthroughs in molecular oncology. However, several critical challenges remain unresolved and urgently demand attention from the research community. In particular, while TNT and NOM have expanded the therapeutic landscape, their long-term oncologic safety, patient selection criteria, and optimal sequencing strategies require further validation in both clinical trials and real-world settings.

One of the most pressing issues is the need to robustly define the long-term oncological safety of NOM. While emerging data suggest that carefully selected patients may achieve durable organ preservation without compromising survival, the risk of local regrowth and its impact on distant recurrence remain inadequately characterized. Large-scale, multicenter, real-world registries dedicated to NOM—incorporating detailed surveillance protocols and longitudinal follow-up—are essential for establishing reliable benchmarks for clinical outcomes, QOL, and cost-effectiveness. Furthermore, harmonizing definitions of cCR and developing consensus-driven surveillance algorithms will be critical for global comparability and adoption.

Another area of critical importance is the optimization of TNT sequencing to achieve sustained therapeutic response while minimizing toxicity. Although both induction and consolidation approaches have demonstrated benefits in randomized trials, direct comparisons are limited, and predictive factors for individualizing sequence selection have not yet been established. Adaptive trial designs and biomarker-guided stratification should be prioritized in future studies to address patient heterogeneity and facilitate the rapid integration of novel agents, including immunotherapies and molecular-targeted drugs, into neoadjuvant regimens.

The most significant transformative potential lies in the development and clinical implementation of validated predictive biomarkers. The integration of multi-omics platforms, next-generation sequencing, and advanced radiomics holds promise for precise patient stratification and real-time response assessment. However, the translation of these technologies into actionable clinical tools is hindered by a lack of prospective validation and standardization. Multi-institutional consortia and international collaborative efforts are warranted to accelerate biomarker discovery and validation.

Finally, the advent of artificial intelligence (AI) offers unprecedented opportunities for data-driven, personalized decision-making. AI-assisted diagnostic tools, predictive analytics, and digital pathology platforms are poised to enhance risk stratification, optimize treatment selection, and reduce inter-observer variability. Future research should focus on integrating AI with clinical, molecular, and imaging data within prospective trials and registry studies.

In conclusion, the next era of LARC management will be defined by multidisciplinary collaboration, patient-centered innovation, and the rigorous pursuit of precision oncology. Addressing the outlined challenges through adaptive research strategies and technological integration will be essential to realizing the full potential of neoadjuvant and organ-preserving approaches, thereby improving survival and QOL for patients worldwide.

## Figures and Tables

**Table 1 cancers-17-02540-t001:** Outcomes of clinical trials on preoperative RT/CRT for LARC.

Trial Name	Period	N	Treatment Arm	pCR	LR	OS	DFS
Swedish [31,32]	1987–1990	1168	pre-operative SC-RT	NA	* 9% (13 years)	* 38% (13 years)	* 72% (CSS)
			Surgery alone	NA	26% (13 years)	30% (13 years)	62% (CSS)
Dutch trial [6,33,34]	1996–1999	1861	pre-operative SC-RT	NA	* 5% (10 years)	48% (10 years)	NA
			Surgery alone	NA	11% (10 years)	49% (10 years)	NA
EORTC 22921 [35,36,37]	1993–2003	1011	pre-operative LC-RT	5.3%	22.4% (10 years)	49.4% (10 years)	44.2% (10 years)
			pre-operative LC-RT + adj CT		* 14.5% (10 years)		
			pre-operative LC-CRT	* 13.7%	* 11.8% (10 years)	50.7% (10 years)	46.4% (10 years)
			pre-operative LC-CRT + adj CT		* 11.7% (10 years)		
FFCD 9203 [38]	1993–2003	762	pre-operative LC-RT	3.6%	16.5% (5 years)	67.9% (5 years)	55.5% (5 years)
			pre-operative LC-CRT	* 11.4%	* 8.1% (5 years)	67.4% (5 years)	59.4% (5 years)
Polish trial [39]	1999–2002	312	pre-operative SC-RT	0.7%	10.6% (4 years)	67.2% (4 years)	58.4% (4 years)
			pre-operative LC-CRT	* 16.1%	15.6% (4 years)	66.2% (4 years)	55.6% (4 years)
TROG 01.04 [40]	2001–2006	326	pre-operative SC-RT	1%	7.5% (3 years)	74% (5 years)	NA
			pre-operative LC-CRT	* 15.1%	4.4% (3 years)	70% (5 years)	NA
CAO/ARO/AIO-94 [7,41]	1994–2002	823	pre-operative LC-CRT	8%	* 7.1% (10 years)	59.6% (10 years)	68.1% (10 years)
			post-operative LC-CRT	-	10.1% (10 years)	59.9% (10 years)	67.8% (10 years)
MRC CR07 and NCIC-CTG C016 [5,42]	1998–2005	1350	pre-operative SC-RT	NA	* 4.4% (3 years)	80.3% (3 years)	* 77.5% (3 years)
			post-operative LC-CRT	-	10.6% (3 years)	78.6% (3 years)	71.5% (3 years)
NSASBP-R04 [43,44]	2004–2010	1608	pre-operative LC-CRT(5FU)	17.8%	11.2% (3 years)	79.9% (5 years)	66.4% (5 years)
			pre-operative LC-CRT(Cape)	20.0%	11.8% (3 years)	80.8% (5 years)	67.7% (5 years)
			pre-operative LC-CRT (+Ox)	19.5%	11.2% (3 years)	81.3% (5 years)	69.2% (5 years)
			pre-operative LC-CRT (−Ox)	17.8%	12.1% (3 years)	79.0% (5 years)	64.2% (5 years)
ACCORD 12 [45,46]	2005–2008	598	pre-operative LC-CRT (−Ox)	13.9%	8.8% (5 years)	73% (5 years)	63.1% (5 years)
			pre-operative LC-CRT (+Ox)	19.2%	7.8% (5 years)	82% (5 years)	66.1% (5 years)
CAO/ARO/AIO-04 [47,48]	2006–2010	1265	pre-operative LC-CRT (−Ox)	13%	4.6% (3 years)	88.0% (3 years)	71.2% (3 years)
			pre-operative LC-CRT (+Ox)	* 17%	2.9% (3 years)	88.7% (3 years)	* 75.9% (5 years)

pCR, pathological complete response; LR, local recurrence; OS, overall survival; DFS, disease-free survival; SC-RT, short-course radiotherapy; LC-RT, long-course radiotherapy; LC-CRT, long-course chemoradiotherapy; NA, not available; CSS, cancer-specific survival; adj CT, adjuvant chemotherapy; 5FU, 5-fluorouracil; Cape, Capecitabine; Ox, Oxaliplatin; *, statistically significant.

**Table 2 cancers-17-02540-t002:** Outcomes of clinical trials on TNT for LARC.

Trial Name	Phase	N	Treatment Arm	pCR	LR/LF	DFS/DRTF	MFS/DM	OS
PRODIGE23 [71,75]	3	461	INCT-TNT (FOLFIRINOX → LC-CRT)	* 28%	5.3% (7 years) ^a^	* 67.6% (7 years) ^c^	* 79.2% (7 years) ^e^	* 81.9% (7 years)
			LC-CRT	12%	8.1% (7 years) ^a^	62.5% (7 years) ^c^	72.3% (7 years) ^e^	76.1% (7 years)
Polish II [70,76]	3	515	CNCT-TNT (SC-RT → FOLFOX4)	16%	35% (8 years) ^b^	43% (8 years) ^c^	36% (8 years) ^f^	49% (8 years)
			LC-CRT	12%	32% (8 years) ^b^	41% (8 years) ^c^	34% (8 years) ^f^	49% (8 years)
RAPIDO [68,77,78]	3	920	CNCT-TNT (SC-RT → CAPOX or FOLFOX4)	* 28%	10.2% (5 years) ^a^	* 27.8% (5 years) ^d^	* 23.0% (5 years) ^f^	81.7% (5 years)
			LC-CRT	14%	* 6.1% (5 years) ^a^	34.0% (5 years) ^d^	30.4% (5 years) ^f^	80.2% (5 years)
STELLAR [69]	3	599	CNCT-TNT (SCRT → CAPOX)	* 21.8%	8.4% (3 years) ^a^	64.5% (3 years) ^c^	77.1% (3 years) ^e^	* 86.5% (3 years)
			LC-CRT	12.3%	11.0% (3 years) ^a^	62.3% (3 years) ^c^	75.3% (3 years) ^e^	75.1% (3 years)
CAO/ARO/AIO-12 [79,80]	2	311	INCT-TNT (FOLFOX → LC-CRT)	17%	6% (3 years) ^a^	73% (3 years) ^c^	18% (3 years) ^f^	92% (3 years)
			CNCT-TNT (LC-CRT → FOLFOX)	* 25%	5% (3 years) ^a^	73% (3 years) ^c^	16% (3 years) ^f^	92% (3 years)
OPRA [81,82]	2	324	INCT-TNT (CAPOX or FOLFOX → LC-CRT) → TME or NOM	NA	6% (5 years) ^a^	71% (5 years) ^c^	80% (5 years) ^e^	88% (5 years)
			CNCT-TNT (LC-CRT → CAPOX or FOLFOX) → TME or NOM	NA	10% (5 years) ^a^	69% (5 years) ^c^	78% (5 years) ^e^	85% (5 years)

TNT, total neoadjuvant therapy; pCR, pathological complete response; LR, local recurrence; LF, local failure; DFS, disease-free survival; DRTF, disease-related treatment failure; MFS, metastasis-free survival; DM, distant metastases; OS, overall survival; INCT, induction chemotherapy; CNCT, consolidation chemotherapy; LC-CRT, long-course chemoradiotherapy; SC-RT, short-course radiotherapy; TME, total mesorectal excision; NA, not available; *, statistically significant; ^a^, LR; ^b^, LF; ^c^, DFS; ^d^, DRTF; ^e^, MFS; ^f^, DM.

**Table 3 cancers-17-02540-t003:** Outcomes of clinical trials on NAC for LARC.

Trial Name	Phase	N	Treatment Arm	pCR	LR	OS	DFS
PROSPECT [92]	2/3	1194	NAC (mFOLFOX6) → Surgery	21.9%	1.8% (5 years)	89.5% (5 years)	80.8% (5 years)
			LC-CRT → Surgery	24.3%	1.6% (5 years)	90.2% (5 years)	78.6% (5 years)
FOWARC [93,94,95]	3	495	LC-CRT (5FU/LV) → Surgery → adj CT (5FU/LV)	14.0%	10.8% (10 years)	65.9% (10 years)	52.5% (10 years)
			LC-CRT (mFOLFOX6) → Surgery → adj CT (mFOLFOX6)	* 27.5%	8.0% (10 years)	72.3% (10 years)	62.6% (10 years)
			NAC (mFOLFOX6) → Surgery → adj CT (mFOLFOX6)	6.6%	9.6% (10 years)	73.4% (10 years)	60.5% (10 years)
CONVERT [96]	3	663	NAC (CAPOX) → Surgery	11.0%	NA	NA	NA
			LC-CRT (Cape) → Surgery	13.8%	NA	NA	NA

NAC, neoadjuvant chemotherapy; pCR, pathological complete response; LR, local recurrence; OS, overall survival; DFS, disease-free survival; LC-CRT, long-course chemoradiotherapy; adj CT, adjuvant chemotherapy; NA, not available; *, statistically significant.

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
