# Peer review of "Neoadjuvant Treatment for Locally Advanced Rectal Cancer: Current Status and Future Directions"

_cancers, 2025, doi:10.3390/cancers17152540_

Round 1
Reviewer 1 Report
Comments and Suggestions for Authors
This narrative review provides a meticulous and extensive examination of neoadjuvant strategies for locally advanced rectal cancer (LARC). The manuscript traverses conventional preoperative radiotherapy (RT) and chemoradiotherapy (CRT), the evolving paradigms of total neoadjuvant therapy (TNT), non-operative management (NOM), and the emerging role of molecular-targeted agents and artificial intelligence. Emphasis is placed on treatment individualization through integration of radiological staging, pathological response data, and potential biomarkers. The review is particularly valuable for summarizing key randomized controlled trials (RCTs) and meta-analyses that have shaped current clinical practice and guidelines.
The manuscript is comprehensive, presenting a well-structured, updated synthesis of the state-of-the-art in preoperative rectal cancer management. Nonetheless, several aspects merit clarification and refinement before acceptance. These range from stylistic and formatting inconsistencies to more substantial issues, such as critical interpretation of conflicting data, the need for deeper engagement with ongoing controversies (e.g., ideal TNT sequencing), and the lack of a future-oriented, hypothesis-generating conclusion. I would recommend a major revision to ensure the paper meets the expectations of an international readership and aligns more closely with the standards of high-impact oncology literature.
- One of the central themes (TNT) is treated in a thorough but predominantly narrative fashion. The discussion heavily leans on summarizing trials like RAPIDO, PRODIGE 23, OPRA, and CAO/ARO/AIO-12, but falls short in interpreting their divergent designs and clinical implications. In particular, the ongoing debate regarding sequencing—induction versus consolidation chemotherapy—is only superficially addressed.
- The manuscript would greatly benefit from a comparative discussion that highlights the rationale, patient selection criteria, and outcomes associated with each approach. For instance, the OPRA trial has fundamentally shifted NOM strategies by associating higher organ preservation rates with consolidation TNT, yet this key nuance is not critically explored. Additionally, the impact of compliance rates, toxicity burden, and tumor regression dynamics as influenced by sequence timing should be better contextualized.
- The discussion of non-operative management is clinically relevant but overly reliant on retrospective series and institutional protocols. It lacks a robust critique of the limitations in current surveillance strategies and the heterogeneity in defining complete clinical response (cCR). There is no mention of ongoing efforts to standardize cCR criteria or the variability in follow-up protocols that compromise the generalizability of the NOM approach. A deeper exploration of the limitations in imaging, digital rectal examination, and endoscopic assessment—especially in the absence of uniform biomarkers—would reinforce the need for further research.
- The sections on molecular-targeted agents and immunotherapies are underdeveloped. While the manuscript lists promising agents and early-phase trials, it fails to critically examine the translational challenges facing their integration into standard neoadjuvant regimens. For instance, the discussion does not address the limited predictive power of current biomarkers, nor does it mention the absence of validated selection criteria for agents such as bevacizumab or checkpoint inhibitors. Furthermore, toxicities, immune-related adverse events, and suboptimal histologic response rates remain underappreciated. The manuscript could be strengthened by identifying the types of molecular or immune profiling that might enable a shift from empiricism to precision-based neoadjuvant care.
- The artificial intelligence and omics section reads as speculative and detached from clinical applicability. While the potential of AI and machine learning in response prediction and image-based staging is acknowledged, the lack of real-world pilot data or ongoing prospective validation trials makes this section weak. If the authors choose to retain it, I recommend citing existing models—such as radiomic signatures that predict pCR after CRT—or AI-assisted pathology tools under investigation. Otherwise, this content should be trimmed or deferred to a dedicated review.
- The language and structure of the paper are mostly clear, though the narrative flow is occasionally interrupted by long tables and repetition of trial acronyms. Some tables could be summarized or integrated within the main text more effectively. Furthermore, terminology should be harmonized—using “pathological complete response (pCR)” consistently, for instance—and reference formatting needs unification across the manuscript.
- What is most notably missing from the manuscript is a truly integrative conclusion. Currently, it reads as a summary rather than a call to action. The conclusion should articulate key unresolved questions—such as the long-term oncologic safety of NOM, the optimal TNT sequence for durable response, and the need for validated predictive biomarkers for treatment selection. Moreover, it should propose a strategic roadmap for future research, including adaptive trial designs, real-world registries for NOM, and AI-assisted decision tools for personalized treatment planning.
Author Response
Response to Reviewer 1
We would like to sincerely thank Reviewer 1 for their thorough reading of our manuscript and for providing constructive and insightful feedback. We appreciate your positive evaluation as well as your valuable suggestions for improvement, all of which have contributed significantly to enhancing the scientific rigor and clarity of our review. Below, we provide a point-by-point response to each of your comments.
Comment 1:
This narrative review provides a meticulous and extensive examination of neoadjuvant strategies for locally advanced rectal cancer (LARC). The manuscript traverses conventional preoperative radiotherapy (RT) and chemoradiotherapy (CRT), the evolving paradigms of total neoadjuvant therapy (TNT), non-operative management (NOM), and the emerging role of molecular-targeted agents and artificial intelligence. Emphasis is placed on treatment individualization through integration of radiological staging, pathological response data, and potential biomarkers. The review is particularly valuable for summarizing key randomized controlled trials (RCTs) and meta-analyses that have shaped current clinical practice and guidelines.
Response:
We are grateful for your positive evaluation and encouraging remarks regarding the comprehensiveness and value of our review.
Comment 2:
The manuscript is comprehensive, presenting a well-structured, updated synthesis of the state-of-the-art in preoperative rectal cancer management. Nonetheless, several aspects merit clarification and refinement before acceptance. These range from stylistic and formatting inconsistencies to more substantial issues, such as critical interpretation of conflicting data, the need for deeper engagement with ongoing controversies (e.g., ideal TNT sequencing), and the lack of a future-oriented, hypothesis-generating conclusion. I would recommend a major revision to ensure the paper meets the expectations of an international readership and aligns more closely with the standards of high-impact oncology literature.
Response:
Thank you for your constructive comments and for identifying specific areas in need of improvement.
We have taken your suggestions seriously and have revised the manuscript accordingly.
We believe that these revisions have addressed your concerns and have further improved the overall quality and impact of our work. Specific changes are detailed below.
Comment 3:
One of the central themes (TNT) is treated in a thorough but predominantly narrative fashion. The discussion heavily leans on summarizing trials like RAPIDO, PRODIGE 23, OPRA, and CAO/ARO/AIO-12, but falls short in interpreting their divergent designs and clinical implications. In particular, the ongoing debate regarding sequencing—induction versus consolidation chemotherapy—is only superficially addressed. The manuscript would greatly benefit from a comparative discussion that highlights the rationale, patient selection criteria, and outcomes associated with each approach.
For instance, the OPRA trial has fundamentally shifted NOM strategies by associating higher organ preservation rates with consolidation TNT, yet this key nuance is not critically explored. Additionally, the impact of compliance rates, toxicity burden, and tumor regression dynamics as influenced by sequence timing should be better contextualized.
Response:
Thank you for this important suggestion.
We have created a new subsection, “5.4. Comparative Analysis and Interpretation of Recent Clinical Trials on TNT” (lines 565–608), in which we provide an in-depth comparative discussion of major TNT trials. This section specifically addresses the rationale, patient selection, and outcomes associated with induction versus consolidation chemotherapy, with particular emphasis on the pivotal role of the OPRA trial in shaping current NOM strategies. We also discuss the impact of compliance rates, toxicity, and tumor regression as influenced by sequencing.
Comment 4:
The discussion of non-operative management is clinically relevant but overly reliant on retrospective series and institutional protocols. It lacks a robust critique of the limitations in current surveillance strategies and the heterogeneity in defining complete clinical response (cCR). There is no mention of ongoing efforts to standardize cCR criteria or the variability in follow-up protocols that compromise the generalizability of the NOM approach. A deeper exploration of the limitations in imaging, digital rectal examination, and endoscopic assessment—especially in the absence of uniform biomarkers—would reinforce the need for further research.
Response:
We appreciate this insightful observation.
While we had previously described the diagnostic methods and common criteria for cCR in section “7.1. Diagnosis of cCR and Selection of Candidates for NOM” (lines 736–760), we recognize the need for a more critical discussion of the heterogeneity in cCR definitions and the lack of standardized surveillance protocols. In response, we have added a new subsection, “9.2. Challenges in cCR Diagnosis and Surveillance Strategies in NOM” (lines 936–980), where we critically address these issues and highlight the limitations of current assessment modalities and follow-up protocols.
Comment 5:
The sections on molecular-targeted agents and immunotherapies are underdeveloped. While the manuscript lists promising agents and early-phase trials, it fails to critically examine the translational challenges facing their integration into standard neoadjuvant regimens. For instance, the discussion does not address the limited predictive power of current biomarkers, nor does it mention the absence of validated selection criteria for agents such as bevacizumab or checkpoint inhibitors. Furthermore, toxicities, immune-related adverse events, and suboptimal histologic response rates remain underappreciated. The manuscript could be strengthened by identifying the types of molecular or immune profiling that might enable a shift from empiricism to precision-based neoadjuvant care.
Response:
Thank you for highlighting this important area.
In response, and also reflecting Reviewer 3’s suggestions, we have reorganized and expanded our discussion on immunotherapies by creating an independent section, “8. Immune Checkpoint Inhibitors for LARC” (lines 839–907).
We have also included discussion of toxicity and adverse events of molecular-targeted therapies at the end of “5.5. Molecular-Targeted Agents within TNT” (lines 656–666).
Furthermore, the current limitations of biomarkers and the need for molecular and immune profiling are discussed in detail in the newly added “9.3. The Need for Improved Patient Stratification and Biomarker-Driven Approaches” (lines 981–1031). Additionally, issues related to molecular classification are addressed in “3.3. Molecular Classification” (lines 145–186).
Comment 6:
The artificial intelligence and omics section reads as speculative and detached from clinical applicability. While the potential of AI and machine learning in response prediction and image-based staging is acknowledged, the lack of real-world pilot data or ongoing prospective validation trials makes this section weak. If the authors choose to retain it, I recommend citing existing models—such as radiomic signatures that predict pCR after CRT—or AI-assisted pathology tools under investigation. Otherwise, this content should be trimmed or deferred to a dedicated review.
Response:
We agree with your assessment.
Given the already substantial scope of this manuscript, we have opted not to include an extensive discussion of AI.
Instead, we briefly mention the role of radiomics in response assessment within “9.2. Challenges in cCR Diagnosis and Surveillance Strategies in NOM” (lines 956–960), and touch on the future potential of AI in “10. Conclusion and Future Directions” (lines 1064–1069).
A more comprehensive analysis of AI applications is deferred to future dedicated reviews.
Comment 7:
The language and structure of the paper are mostly clear, though the narrative flow is occasionally interrupted by long tables and repetition of trial acronyms. Some tables could be summarized or integrated within the main text more effectively. Furthermore, terminology should be harmonized—using “pathological complete response (pCR)” consistently, for instance—and reference formatting needs unification across the manuscript.
Response:
Thank you for these suggestions.
We have revised Table 1 to improve readability by reducing excessive information and simplifying the “treatment arm” column. Table 3 was likewise simplified. We retained essential information in the tables to facilitate comparison of key study results.
With regard to trial acronyms, we believe they aid reader comprehension in a review of this scope, but we have aimed to limit redundancy.
Reference formatting has been thoroughly checked and any errors or inconsistencies have been corrected.
Comment 8:
What is most notably missing from the manuscript is a truly integrative conclusion. Currently, it reads as a summary rather than a call to action. The conclusion should articulate key unresolved questions—such as the long-term oncologic safety of NOM, the optimal TNT sequence for durable response, and the need for validated predictive biomarkers for treatment selection. Moreover, it should propose a strategic roadmap for future research, including adaptive trial designs, real-world registries for NOM, and AI-assisted decision tools for personalized treatment planning.
Response:
Thank you for this valuable suggestion.
In response, we have revised the conclusion and added a dedicated section, “10. Conclusion and Future Directions,” to highlight the main unresolved issues and future research directions as you recommended. (lines 1032–1074)
We appreciate your guidance in strengthening the final section of our manuscript.
Once again, we thank Reviewer 1 for their invaluable feedback, which has greatly contributed to improving the quality, clarity, and scientific impact of our manuscript.
Reviewer 2 Report
Comments and Suggestions for Authors
Thank you for submitting this outstanding class comprehensive review regarding neoadjuvant treatment strategies and future directions for locally advanced rectal cancer. Th article was well written, clear and easy to read. The topic was important and would significantly add to our knowledge.
Author Response
Response to Reviewer 2
We would like to express our sincere gratitude to Reviewer 2 for taking the time to review our manuscript and for your positive and encouraging comments. We are very pleased that you found our review to be clear, well written, and of potential significance to the field. Your supportive evaluation is greatly appreciated and has been a source of motivation for our team.
Thank you once again for your kind feedback.
Reviewer 3 Report
Comments and Suggestions for Authors
Although the review article titled "Neoadjuvant Treatment for Locally Advanced Rectal Cancer:
Current Status and Future Directions" is quite comprehensive and informative, a thorough revision is needed to improve the quality of data presentation.
- The definition of LARC is not provided. The authors should clarify what they mean by LARC in terms of TNM classification.
- The molecular classification scheme for rectal cancer should also be introduced, as the manuscript discusses targeted therapies and immunotherapy agents. This information should be included early in the text—ideally in a new subsection alongside the definition of LARC.
- For unclear reasons, the authors address the diagnosis of NOM in one of the final subsections. This contradicts standard clinical structure, where diagnosis is typically presented prior to treatment.
- Also, the discussion of immunotherapy is placed in the "Future Directions" subsection. It is unclear why this was done, especially given that targeted therapy is discussed in a separate, dedicated subsection. Generally, a "Future Directions" section should focus on identifying gaps in the current evidence and proposing areas for future research, rather than introducing additional treatment options already in use for LARC.
- It is also unclear what value adds the discussion of QoL aspects associated with the provision of NOM.
- Finally, the criteria used for study selection in this review are not described. Considering the large body of literature on this topic, it would be appropriate to include a subsection outlining the methodology of the literature search and study selection process.
Author Response
Response to Reviewer 3
We sincerely thank Reviewer 3 for their careful review of our manuscript and for providing numerous constructive suggestions. We greatly appreciate your efforts and have thoroughly revised our manuscript in response to your feedback. We believe these revisions have helped to address your concerns and have further improved the quality of our review. Our detailed responses to each comment are as follows:
Comment 1:
The definition of LARC is not provided. The authors should clarify what they mean by LARC in terms of TNM classification.
Response:
Thank you for this important point.
In the revised manuscript, we have created a new section, “3. Definition, Diagnosis, and Molecular Classification of LARC,” and now provide a clear definition of LARC, including relevant TNM classification details, in subsection “3.1. Definition of LARC” (lines 90–106).
Comment 2:
The molecular classification scheme for rectal cancer should also be introduced, as the manuscript discusses targeted therapies and immunotherapy agents. This information should be included early in the text—ideally in a new subsection alongside the definition of LARC.
Response:
Thank you for your valuable suggestion.
In accordance with your advice, we have included a detailed description of the molecular classification scheme for rectal cancer in subsection “3.3. Molecular Classification,” within the newly established “3. Definition, Diagnosis, and Molecular Classification of LARC” section (lines 145–186).
Comment 3:
For unclear reasons, the authors address the diagnosis of NOM in one of the final subsections. This contradicts standard clinical structure, where diagnosis is typically presented prior to treatment.
Response:
Thank you for pointing this out.
We have revised the structure of the “7. Non-operative management (NOM)” section so that “7.1. Diagnosis of cCR and Selection of Candidates for NOM” now precedes “7.2. Outcomes of NOM” (lines 736 and 761), reflecting the standard clinical flow.
Comment 4:
Also, the discussion of immunotherapy is placed in the "Future Directions" subsection. It is unclear why this was done, especially given that targeted therapy is discussed in a separate, dedicated subsection. Generally, a "Future Directions" section should focus on identifying gaps in the current evidence and proposing areas for future research, rather than introducing additional treatment options already in use for LARC.
Response:
Thank you for this helpful observation.
In the revised manuscript, we have moved the discussion of immunotherapy to a newly created independent section, “8. Immune Checkpoint Inhibitors for LARC” (lines 839–907).
The former “Future Directions” section has been removed, with relevant content redistributed to “9. Unresolved Issues in Preoperative Treatment for LARC” and “10. Conclusion and Future Directions” (lines 908–1074).
Comment 5:
It is also unclear what value adds the discussion of QoL aspects associated with the provision of NOM.
Response:
Thank you for raising this important point.
As we discuss throughout the manuscript, oncological outcomes are certainly of primary importance in the treatment of LARC; however, the preservation of postoperative quality of life (QoL) is also a critical consideration.
NOM offers the possibility for patients to avoid surgery and thereby reduce the risk of postoperative functional decline. On the other hand, achieving cCR often requires intensive radiotherapy and/or chemotherapy, which can result in adverse events and long-term sequelae similar to those observed after surgery. As noted in our manuscript, although patients managed with NOM generally experience better QoL compared to those undergoing surgery, evidence indicates that around 25% of patients still suffer from severe bowel dysfunction at 2 years, and issues such as fecal incontinence or urinary/sexual dysfunction remain relatively common (see lines 820–824). We believe these data are highly relevant both for selecting appropriate candidates for NOM and for providing patients with balanced and accurate information during follow-up (lines 833–838).
Comment 6:
Finally, the criteria used for study selection in this review are not described. Considering the large body of literature on this topic, it would be appropriate to include a subsection outlining the methodology of the literature search and study selection process.
Response:
Thank you for highlighting this omission.
We have added a new subsection, “2. Literature Search Strategy,” to clearly describe our literature search methodology and study selection process (lines 68–89).
Once again, we thank you for your insightful feedback and constructive criticism, which have been invaluable in improving our manuscript.
Reviewer 4 Report
Comments and Suggestions for Authors
I really enjoyed reading the review “Neoadjuvant Treatment for Locally Advanced Rectal Cancer: Current Status and Future Directions” and I think it's very good.
One point seems important to me, which is not described in enough detail: The classification/definition of ‘advanced rectal cancer’. Please add the following: problems of classification in imaging, the uncertainties of cT and especially cN, clinical overstaging, the superiority of MRI over CT, the importance of the distance of the tumour from the mesorectal fascia (MRF).
Author Response
Response to Reviewer 4
We would like to express our sincere gratitude to Reviewer 4 for reviewing our manuscript and for your positive evaluation. We greatly appreciate your encouraging comments, which have been a strong motivation for us.
Comment:
One point seems important to me, which is not described in enough detail: The classification/definition of ‘advanced rectal cancer’. Please add the following: problems of classification in imaging, the uncertainties of cT and especially cN, clinical overstaging, the superiority of MRI over CT, the importance of the distance of the tumour from the mesorectal fascia (MRF).
Response:
Thank you very much for this important and constructive suggestion.
In response, we have newly added detailed descriptions on these points in section “3. Definition, Diagnosis, and Molecular Classification of LARC,” specifically in “3.1. Definition of LARC” and “3.2. Diagnosis of LARC” (lines 90–144), as you recommended.
We believe these revisions have further improved the clarity and completeness of our manuscript.
Thank you again for your thoughtful feedback and support.
Round 2
Reviewer 1 Report
Comments and Suggestions for Authors
The Authors revised in a very outstanding their paper that has been notably improved and now it is suitable for publication
Reviewer 3 Report
Comments and Suggestions for Authors
Well done!